# Investigating phenotypes of pulmonary COVID-19 recovery: A longitudinal observational prospective multicenter trial

**Thomas Sonnweber[1†], Piotr Tymoszuk[1†‡], Sabina Sahanic[1], Anna Boehm[1], Alex Pizzini[1], Anna Luger[2], Christoph Schwabl[2], Manfred Nairz[1], Philipp Grubwieser[1], Katharina Kurz[1], Sabine Koppelstätter[1], Magdalena Aichner[1], Bernhard Puchner[3], Alexander Egger[4], Gregor Hoermann[4,5], Ewald Wöll[6], Günter Weiss[1], Gerlig Widmann[2], Ivan Tancevski[1]\*, Judith Löffler-Ragg[1]\***

[1]Department of Internal Medicine II, Medical University of Innsbruck, Innsbruck, Austria; [2]Department of Radiology, Medical University of Innsbruck, Innsbruck, Austria; [3]The Karl Landsteiner Institute, Muenster, Austria; [4]Central Institute of Medical and Chemical Laboratory Diagnostics, University Hospital Innsbruck, Innsbruck, Austria; [5]Munich Leukemia Laboratory, Munich, Germany; [6]Department of Internal Medicine, St. Vinzenz Hospital, Zams, Austria

**\*For correspondence:**
Ivan.Tancevski@i-med.ac.at (IT);
Judith.Loeffler@i-med.ac.at
(JL-R)

[†]These authors contributed equally to this work

**Present address:** [‡]Data Analytics As a Service Tirol, Innsbruck, Austria

## Abstract

**Background:** The optimal procedures to prevent, identify, monitor, and treat long-term pulmonary sequelae of COVID-19 are elusive. Here, we characterized the kinetics of respiratory and symptom recovery following COVID-19.

**Methods:** We conducted a longitudinal, multicenter observational study in ambulatory and hospitalized COVID-19 patients recruited in early 2020 (n = 145). Pulmonary computed tomography (CT) and lung function (LF) readouts, symptom prevalence, and clinical and laboratory parameters were collected during acute COVID-19 and at 60, 100, and 180 days follow-up visits. Recovery kinetics and risk factors were investigated by logistic regression. Classification of clinical features and participants was accomplished by unsupervised and semi-supervised multiparameter clustering and machine learning.

**Results:** At the 6-month follow-up, 49% of participants reported persistent symptoms. The frequency of structural lung CT abnormalities ranged from 18% in the mild outpatient cases to 76% in the intensive care unit (ICU) convalescents. Prevalence of impaired LF ranged from 14% in the mild outpatient cases to 50% in the ICU survivors. Incomplete radiological lung recovery was associated with increased anti-S1/S2 antibody titer, IL-6, and CRP levels at the early follow-up. We demonstrated that the risk of perturbed pulmonary recovery could be robustly estimated at early follow-up by clustering and machine learning classifiers employing solely non-CT and non-LF parameters.

**Conclusions:** The severity of acute COVID-19 and protracted systemic inflammation is strongly linked to persistent structural and functional lung abnormality. Automated screening of multiparameter health record data may assist in the prediction of incomplete pulmonary recovery and optimize COVID-19 follow-up management.

**Funding:** The State of Tyrol (GZ 71934), Boehringer Ingelheim/Investigator initiated study (IIS 1199-0424).

**Clinical trial number:**
ClinicalTrials.gov: NCT04416100

## Editor's evaluation

This is an informative paper describing the incidence and predictors of long-term radiological and functional lung abnormalities following COVID-19. Congratulations on the importance of the work!

## Introduction

The ongoing COVID-19 pandemic challenges health-care systems. As of December 2021, the John Hopkins dashboard (*Dong et al., 2020*) reports 276 million cases and 5.4 million COVID-19-related deaths worldwide (*Johns Hopkins Coronavirus Resource Center, 2021*). Although the vast majority of COVID-19 patients display mild disease, approximately 10–15% of cases progress to a severe condition and approximately 5% suffer from critical illness (*Perez-Saez, 2021*; *Huang et al., 2020*). Similar to severe acute respiratory syndrome (SARS) (*Hui et al., 2005*; *Ng et al., 2004*; *Ngai et al., 2010*; *Lam et al., 2009*), a significant portion of COVID-19 patients report lingering or recurring clinical impairment and cardiopulmonary recovery may take several months to years (*Sonnweber et al., 2021*; *Sahanic et al., 2021*; *Caruso et al., 2021*; *Huang et al., 2021b*; *Huang et al., 2021a*; *Faverio et al., 2021*; *Hellemons et al., 2021*; *Zhou et al., 2021*; *Venkatesan, 2021*). This observation has led to the introduction of the term 'long COVID,' defined by the persistence of COVID-19 symptoms for more than 4 weeks, and the 'post-acute sequelae of COVID-19' (PASC) referring to symptom persistence for more than 12 weeks (*Sahanic et al., 2021*; *Shah et al., 2021*; *Sudre et al., 2021b*). Evidence-based strategies for prediction, monitoring, and treatment of PASC are urgently needed (*Raghu and Wilson, 2020*).

We herein prospectively analyzed the prevalence of nonresolving structural and functional lung abnormalities and persistent COVID-19-related symptoms 6 months after diagnosis. Using univariate risk modeling as well as multiparameter clustering and machine learning (ML), we investigated sets of risk factors and tested the operability of ML classifiers at predicting protracted lung and symptom recovery. The classification and prediction procedures were implemented in an open-source risk assessment tool (https://im2-ibk.shinyapps.io/CovILD/).

## Methods
### Study design

The CovILD ('Development of interstitial lung disease in COVID-19') multicenter, longitudinal observational study (*Sonnweber et al., 2021*) was initiated in April 2020. Adult residents of Tyrol, Austria, with symptomatic, PCR-confirmed SARS-CoV-2 infection (*WHO, 2021*) were enrolled by the Department of Internal Medicine II at the Medical University of Innsbruck (primary follow-up center), St. Vinzenz Hospital in Zams, and the acute rehabilitation facility in Münster (*Table 1*). The participants were diagnosed with COVID-19 between 3 March and 29 June 2020. In course of the study, including the 2020 SARS-CoV-2 outbreak and follow-up visits, the regional

**Table 1.** Characteristics of the study population.

| Characteristics (% cohort) | |
|---|---|
| Total participants – no. | 145 |
| Mean age, years | 57.3 (SD = 14.3) |
| Female sex | 42.4% (n = 63) |
| Obesity (body mass index >30 kg/m$^2$) | 19.3% (n = 28) |
| Ex-smoker | 39.3% (n = 57) |
| Active smoker | 2.8% (n = 4) |
| **Acute COVID-19 severity (% cohort)** | |
| Mild: outpatient | 24.8% (n = 36) |
| Moderate: inpatient without oxygen therapy | 25.5% (n = 37) |
| Severe: inpatient with oxygen therapy | 27.6% (n = 40) |
| Critical: intensive care unit | 22.1% (n = 32) |
| **Comorbidities (% cohort)** | |
| None | 22.8% (n = 33) |
| Cardiovascular disease | 40% (n = 58) |
| Pulmonary disease | 18.6% (n = 27) |
| Metabolic disease | 43.4% (n = 63) |
| Chronic kidney disease | 6.9% (n = 10) |
| Gastrointestinal tract diseases | 13.8% (n = 20) |
| Malignancy | 11.7% (n = 17) |

**Table 2.** Hospitalization and medication during acute COVID-19.

| Parameter | Outpatient (n = 36) | Hospitalized (n = 37) | Hospitalized oxygen therapy (n = 40) | Hospitalized intensive care unit (n = 32) |
|---|---|---|---|---|
| Mean hospitalization time, days | 0 (SD = 0) | 6.9 (SD = 3.6) | 11.8 (SD = 6.3) | 34.8 (SD = 15.7) |
| Hospitalized >7 days | 0% (n = 0) | 43.2% (n = 16) | 80% (n = 32) | 100% (n = 32) |
| Anti-infectives | 11.1% (n = 4) | 45.9% (n = 17) | 72.5% (n = 29) | 87.5% (n = 28) |
| Antiplatelet drugs | 2.8% (n = 1) | 10.8% (n = 4) | 22.5% (n = 9) | 25% (n = 8) |
| Anticoagulatives | 2.8% (n = 1) | 2.7% (n = 1) | 5% (n = 2) | 15.6% (n = 5) |
| Corticosteroids*† | 2.8% (n = 1) | 5.4% (n = 2) | 22.5% (n = 9) | 40.6% (n = 13) |
| Immunosuppression‡† | 0% (n = 0) | 2.7% (n = 1) | 5% (n = 2) | 9.4% (n = 3) |

*From the week 4 post diagnosis on, at the discretion of the physician.
†Subsumed under 'immunosuppression, acute COVID-19' for data analysis.
‡Immunosuppressive medication prior to COVID-19.

health system was able to guarantee an unrestricted, optimal standard of diagnostics and care for all participants. Corticosteroids were not standard of care during the recruitment period of the study, thus were not administered as a therapy of acute COVID-19. Some participants with nonresolving pneumonia received systemic steroids beginning from week 4 post diagnosis at the discretion of the physician (*Table 2*). The analysis endpoints were the presence of any, mild (severity score ≤ 5), and moderate-to-severe (severity score > 5) lung computed tomography (CT) abnormalities, impaired lung function (LF), and persistent COVID-19 symptoms at the 180-day follow-up visit (*Table 3*).

In total, 190 COVID-19 patients were screened for participation. Thereof, n = 18 subjects refused to give informed consent, n = 27 declared difficulties to appear at the study follow-ups. Data of n = 145 participants were eligible for analysis (*Figure 1*). All participants gave written informed consent. The study was approved by the Institutional Review Board at the Medical University of Innsbruck (approval number: 1103/2020) and registered at ClinicalTrials.gov (NCT04416100).

## Procedures

We retrospectively assessed patient characteristics during acute COVID-19 and performed follow-up investigations at 60 days (63 ± 23 days [mean ± SD]; visit 1), 100 days (103 ± 21 days; visit 2), and 180 days (190 ± 15 days; visit 3) after diagnosis of COVID-19. Each visit included symptom and physical performance assessment with a standardized questionnaire, LF testing, standard laboratory testing, and a CT scan of the chest. The variables available for analysis with their stratification schemes are listed in *Appendix 1—table 1*.

Serological markers were determined in certified laboratories (Central Institute of Clinical and Chemical Laboratory Diagnostics, Rheumatology and Infectious Diseases Laboratory, both at the University Hospital of Innsbruck). C-reactive protein (CRP), interleukin-6 (IL-6), N-terminal pro natriuretic peptide (NT-proBNP), and serum ferritin were measured using a Roche Cobas 8000 analyzer. D-dimer was determined with a Siemens BCS-XP instrument using the Siemens D-Dimer Innovance reagent. Anti-S1/S2 protein SARS-CoV-2 immunoglobulin gamma (IgG) were quantified with LIAISON

**Table 3.** Radiological, functional, and clinical study outcomes.

| Outcome | 60-day follow-up | 100-day follow-up | 180-day follow-up |
|---|---|---|---|
| Any lung CT abnormalities (complete: n = 103) | 74.8% (n = 77) | 60.2% (n = 62) | 48.5% (n = 50) |
| Mild lung CT abnormalities (severity score ≤ 5) (complete: n = 103) | 26.2% (n = 27) | 36.9% (n = 38) | 29.1% (n = 30) |
| Moderate-to-severe CT abnormalities (severity score > 5) (complete: n = 103) | 48.5% (n = 50) | 23.3% (n = 24) | 19.4% (n = 20) |
| Functional lung impairment (complete: n = 116) | 39.7% (n = 46) | 37.1% (n = 43) | 33.6% (n = 39) |
| Persistent symptoms (complete: n = 145) | 79.3% (n = 115) | 67.6% (n = 98) | 49% (n = 71) |

CT = computed tomography.

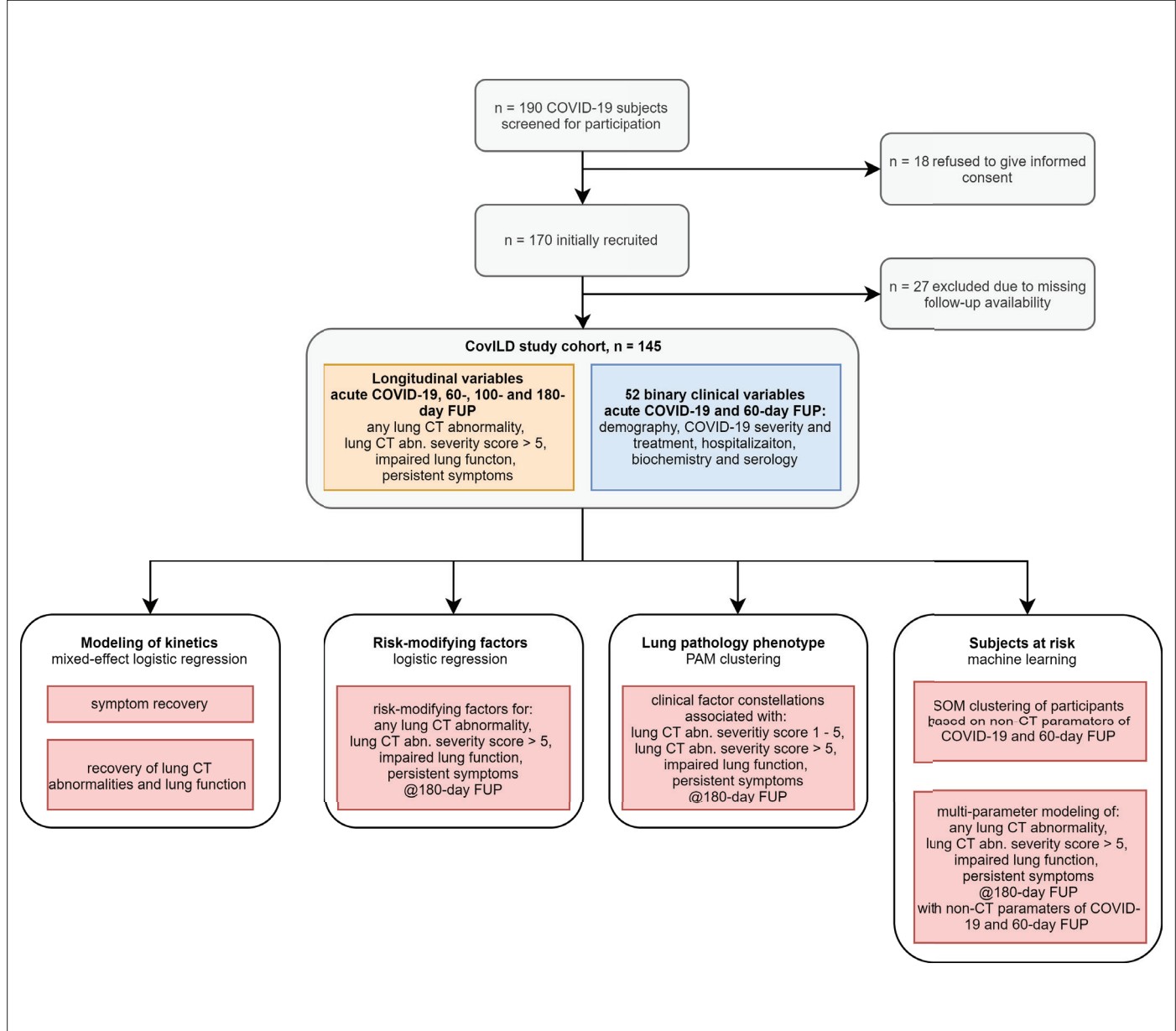

**Figure 1.** Study inclusion flow diagram and analysis scheme.

chemoluminescence assay (DiaSorin, Italy), expressed as binding antibody units (BAU, conversion factor = 5.7) and stratified by quartiles (*Ferrari et al., 2021*).

Low-dose (100 kVp tube potential) craniocaudal CT scans of the chest were acquired without iodine contrast and without ECG gating on a 128-slice multidetector CT (128 × 0.6 mm collimation, 1.1 spiral pitch factor, SOMATOM Definition Flash, Siemens Healthineers, Erlangen, Germany). In case of clinically suspected pulmonary embolism, CT scans were performed with a contrast agent. Axial reconstructions were done with 1 mm slices. CT scans were evaluated for ground-glass opacities, consolidations, bronchial dilation, and reticulations as defined by the Fleischner Society. Lung findings were graded with a semi-quantitative CT severity score (0–25 points) (*Sonnweber et al., 2021*).

Impaired LF was defined as (1) forced vital capacity (FVC) < 80% or (2) forced expiratory volume in 1 s ($FEV_1$) < 80%, or (3) $FEV_1$:FVC < 70% or (4) total lung capacity (TLC) < 80% or (5) diffusing capacity of carbon monoxide (DLCO) < 80% predicted.

## Statistical analysis

Statistical analyses were performed with R version 4.0.5 (*Figure 1*). Data transformation and visualization were accomplished by *tidyverse* (*Wickham et al., 2019*), ggplot2 (*Wickham, 2016*), *ggvenn*, *plotROC* (*Sachs, 2017*), and *cowplot* (*Wilke, 2019*) packages. The recorded variables were binarized as shown in *Appendix 1—table 1*. Acute COVID-19 severity strata were defined as presented in *Table 1*. p-Values were corrected for multiple comparisons with the Benjamini–Hochberg method (*Benjamini and Hochberg, 1995*), and effects were termed significant for p<0.05.

## Variable overlap, kinetics, and risk modeling

Overlap between the 180-day follow-up outcome features was assessed by analysis of quasi-proportional Venn plots (package *nVennR*) (*Pérez-Silva et al., 2018*) and calculation of the Cohen's κ statistic (package *vcd*) (*Fleiss et al., 1969*). Kinetics of binary outcome variables in participants subsets with the complete longitudinal data record was modeled with mixed-effect logistic regression (random effect: individual, fixed effect: time, packages *lme4* [*Bates et al., 2015*] and *lmerTest* [*Kuznetsova et al., 2017*]). Analyses in the severity groups were done with separate models. Significance was assessed by the likelihood ratio test (LRT) against the random-term-only model. Univariate risk modeling was performed with fixed-effect logistic regression (*Appendix 1—table 2*). Odds ratio (OR) significance was determined by Wald Z test. In-house-developed linear modeling wrappers around base R tools are available at https://github.com/PiotrTymoszuk/lmqc.

## Cluster analysis

Clustering of non-CT and non-LF binary clinical features (*Appendix 1—table 1*) was accomplished with PAM algorithm (partitioning around medoids, package *cluster*) (*Amato et al., 2019*) and simple matching distance (SMD, package *nomclust*) (*Boriah et al., 2008*). Association analysis for the participants was performed with a combined procedure involving clustering of the observations by the self-organizing map algorithm (SOM, 4 × 4 hexagonal grid, SMD distance, *kohonen* package), followed by clustering of the SOM nodes by the Ward.D2 hierarchical clustering algorithm (Euclidean distance, *hclust*() function, package *stats*) (*Vesanto and Alhoniemi, 2000*; *Kohonen, 1995*; *Wehrens and Kruisselbrink, 2018*). Clustering analyses were performed in the participant subset with the complete set of clustering variables. The selection of the optimal clustering algorithm was motivated by the highest ratio of between-cluster to total variance and the best stability measured by mean classification error in 20-fold cross-validation (CV) (*Figure 6—figure supplement 1A and B*, *Figure 7—figure supplement 1A and B*; *Lange et al., 2004*). The optimal cluster number was determined by the bend of the within-cluster sum-of-squares curve (function *fviz_nbclust*(), package *factoextra*) and by the stability in 20-fold CV (*Figure 6—figure supplement 1C and D*, *Figure 7—figure supplement 1D and F*; *Lange et al., 2004*; *Wang, 2010*), as well as by a visual inspection of the SOM node clustering dendrograms (*Figure 7—figure supplement 1E*). Assignment of 180-day follow-up outcome features to the clusters of clinical parameters was accomplished with a k-nearest neighbor (kNN) label propagation algorithm (*Appendix 1—table 3*; *Sahanic et al., 2021*; *Leng et al., 2013*). Cluster assignment visualization in a four-dimensional principal analysis score plot was done with the *PCAproj*() tool (package *pcaPP*) (*Croux et al., 2007*). To determine the importance of particular clustering variables, the variance (between-cluster to total variance ratio) between the initial cluster structure and the structure with random resampling of the variable was compared, as initially proposed for the random forests ML classifier (*Breiman, 2001*). Frequencies of the outcome events in the participant clusters were compared with $\chi^2$ test. In-house-developed association analysis wrappers are available at https://github.com/PiotrTymoszuk/clustering-tools-2.

## Machine learning

ML classifiers C5.0 (package *C50*) (*Quinlan, 1993*), random forests (*randomForest*) (*Breiman, 2001*), support vector machines with radial kernel (*kernlab*) (*Weston and Watkins, 1998*), neural networks (*nnet*) (*Ripley, 2014*), and elastic net (*glmnet*) (*Friedman et al., 2010*) were trained to predict the 180-day follow-up outcomes employing non-CT and non-LF binary explanatory features (*Appendix 1—table 1*). The ML training was performed in the participant subsets with the complete set of explanatory and outcome variables. The training, optimization, and CV (20-fold, five repetitions) were accomplished by the *train*() tool from *caret* package, with the Cohen's κ statistic as a model selection

metric (*Appendix 1—table 4*; *Kuhn, 2008*). Classifier ensembles were constructed with the elastic net procedure (*caretStack*() function, *caretEnsemble* package, *Appendix 1—table 4*; *Deane-Mayer and Knowles, 2019*). Classifier performance in the training cohort and CV was assessed by receiver-operating characteristics (ROCs), Cohen's κ and accuracy (packages *caret* and *vcd*, *Appendix 1—table 5*; *Fleiss et al., 1969*; *Kuhn, 2008*). Variable importance measures were extracted from the C5.0 (percent variable usage, *c5imp*() function, package *C50*) (*Quinlan, 1993*), random forests (Δ Gini index, *importance*(), package *randomForest*) (*Breiman, 2001*), and elastic net classifiers (regression coefficient β, *coef*(), package *glmnet*) (*Friedman et al., 2010*).

### Pulmonary recovery assessment app

Participant clustering and ML classifiers trained in the CovILD cohort were implemented in an open-source online pulmonary assessment R shiny app (https://im2-ibk.shinyapps.io/CovILD/; code: https://github.com/PiotrTymoszuk/COVILD-recovery-assessment-app). Prediction of the cluster assignment based on the user-provided patient data is done by the kNN label propagation algorithm (*Sahanic et al., 2021*; *Leng et al., 2013*).

## Results

### Patient characteristics

The CovILD study participants (n = 145) were predominantly male (57.8%), age ranging between 19 and 87 years. 77.2% of participants displayed preexisting comorbidity, predominantly cardiovascular and metabolic disease. The cohort included mild (outpatient care, 24.8%), moderate (hospitalization without oxygen supply, 25.5%), severe (hospitalization with oxygen supply, 27.6%), and critical (intensive care unit [ICU] treatment, 22.1%) cases of acute COVID-19 (*Table 1*). The majority of hospitalized participants received anti-infectives during acute COVID-19, anticoagulative, and/or antiplatelet treatment introduced primarily in the ventilated patients. Systemic steroid administration was initiated at the discretion of the physician beginning from week 4 after diagnosis (*Table 2*).

### Clinical recovery after COVID-19

Most patients, irrespective of the acute COVID-19 severity, showed a significant resolution of disease symptoms over time (*Figure 1*, *Figure 2A*). Persistent complaints at the 6-month follow-up were reported by 49% of the study subjects (*Table 3*), with self-reported impaired physical performance (34.7%), sleep disorders (27.1%), and exertional dyspnea (22.8%) as leading manifestations. The frequency of all investigated symptoms declined significantly, even though the pace of their resolution was remarkably slower in the late (100- and 180-day follow-ups) than in the early recovery phase (acute COVID-19 till 60-day follow-up) (*Figure 2B*).

Impaired LF was observed in 33.6% of the participants at the 6-month follow-up (*Table 3*). Except for the critical COVID-19 survivors (60 days: 66.7%; 180 days post-COVID-19: 50%), no significant reduction in the frequency of LF impairment over time was observed (*Figure 3*). At the 6-month follow-up, structural lung abnormalities were found in 48.5% of patients and moderate-to-severe radiological lung alterations (CT severity score > 5) were present in 19.4% of participants (*Table 3*). The majority of the participants with impaired LF displayed radiological lung findings. However, a substantial fraction of CT abnormalities, especially mild ones, were accompanied neither by persistent symptoms nor by LF deficits (*Figure 3—figure supplement 1*, *Figure 3—figure supplement 2*, *Figure 3—figure supplement 3A*).

The frequency, scoring, and recovery of CT lung findings were related to the severity of acute infection. Pulmonary lesions scored > 5 CT severity points at the 180-day follow-up were most frequent in the individuals with severe and critical acute COVID-19 (*Figure 3—figure supplement 3*). Notably, the hospitalized group with oxygen therapy demonstrated the fastest recovery kinetics. As for the symptom resolution, LF and CT lung recovery decelerated in the late phase of COVID-19 convalescence (*Figure 3*).

### Risk factors of protracted recovery

To identify risk factors of delayed recovery at the 6-month follow-up, we screened a set of 52 binary clinical parameters (*Appendix 1—table 1*) recorded during acute COVID-19 and at the 60-day visit

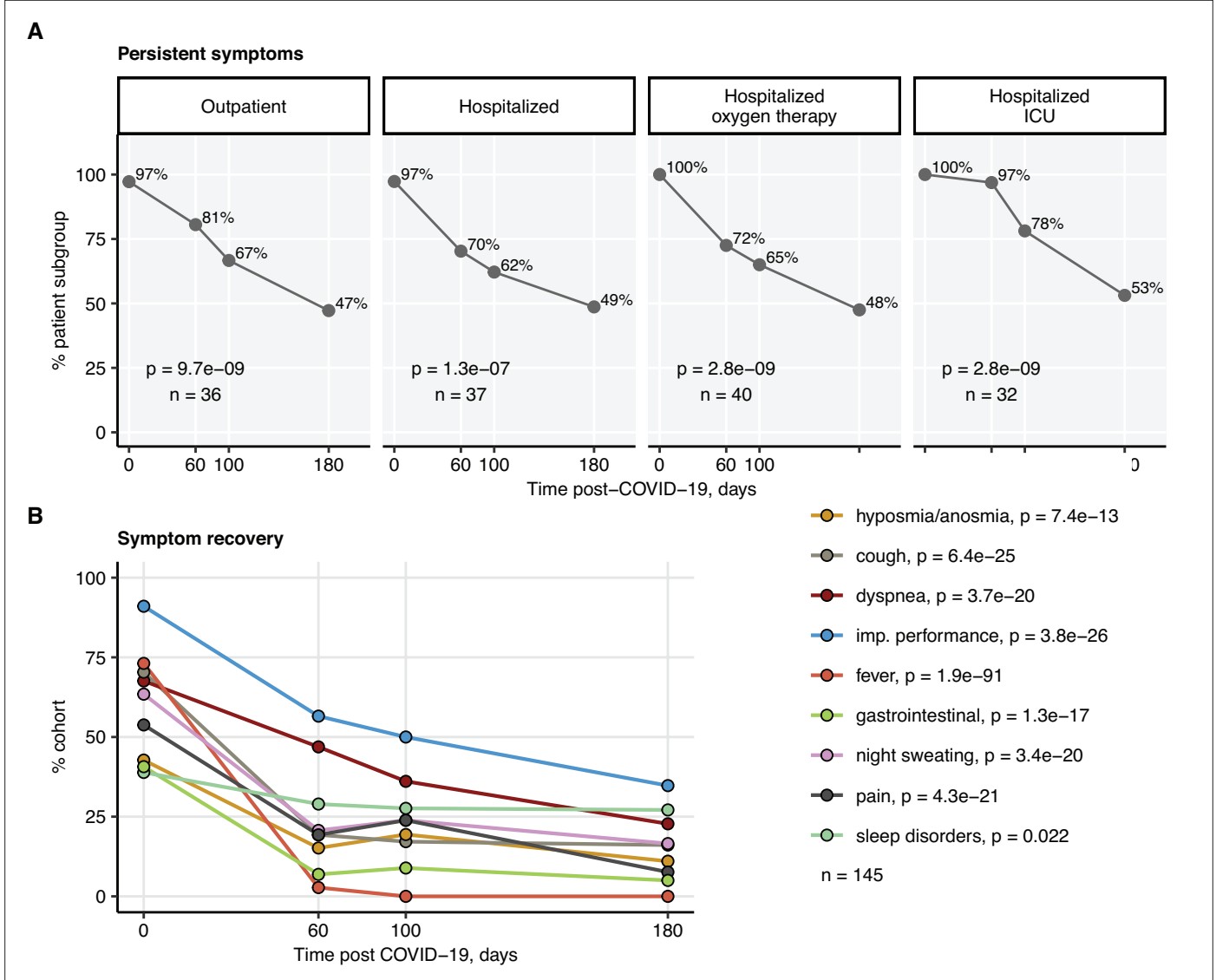

**Figure 2.** Kinetic of recovery from COVID-19 symptoms. Recovery from any COVID-19 symptoms was investigated by mixed-effect logistic modeling (random effect: individual; fixed effect: time). Significance was determined by the likelihood ratio test corrected for multiple testing with the Benjamini–Hochberg method, and p-values and the numbers of complete observations are indicated in the plots. (**A**) Frequencies of individuals with any symptoms in the study cohort stratified by acute COVID-19 severity. (**B**) Frequencies of participants with particular symptoms. imp.: impaired.

by univariate modeling (**Appendix 1—table 2**). By this means, no significant correlates for long-term symptom persistence were identified. Risk factors and readouts of severe and critical COVID-19 including multimorbidity, malignancy, male sex, prolonged hospitalization, ICU stay, and immuno-suppressive therapy were significantly associated with persistent CT (**Figure 4**) and LF abnormalities (**Figure 5**). Persistently elevated inflammatory markers, IL-6 (>7 ng/L) and CRP (>0.5 mg/L), were strong unfavorable risk factors for incomplete radiological and functional pulmonary recovery. Additionally, the biochemical readout of microvascular inflammation, D-dimer (>500 pg/mL) was significantly linked to LF deficits. Low serum anti-S1/S2 IgG titers at the 60-day follow-up and ambulatory acute COVID-19 correlated with an improved pulmonary recovery (**Figures 4 and 5**).

## Clusters of clinical features linked to persistent symptoms and lung abnormalities

Employing the unsupervised PAM algorithm (**Amato et al., 2019**), three clusters of co-occurring non-CT and non-LF clinical features of acute COVID-19 and early convalescence (**Appendix 1—table**

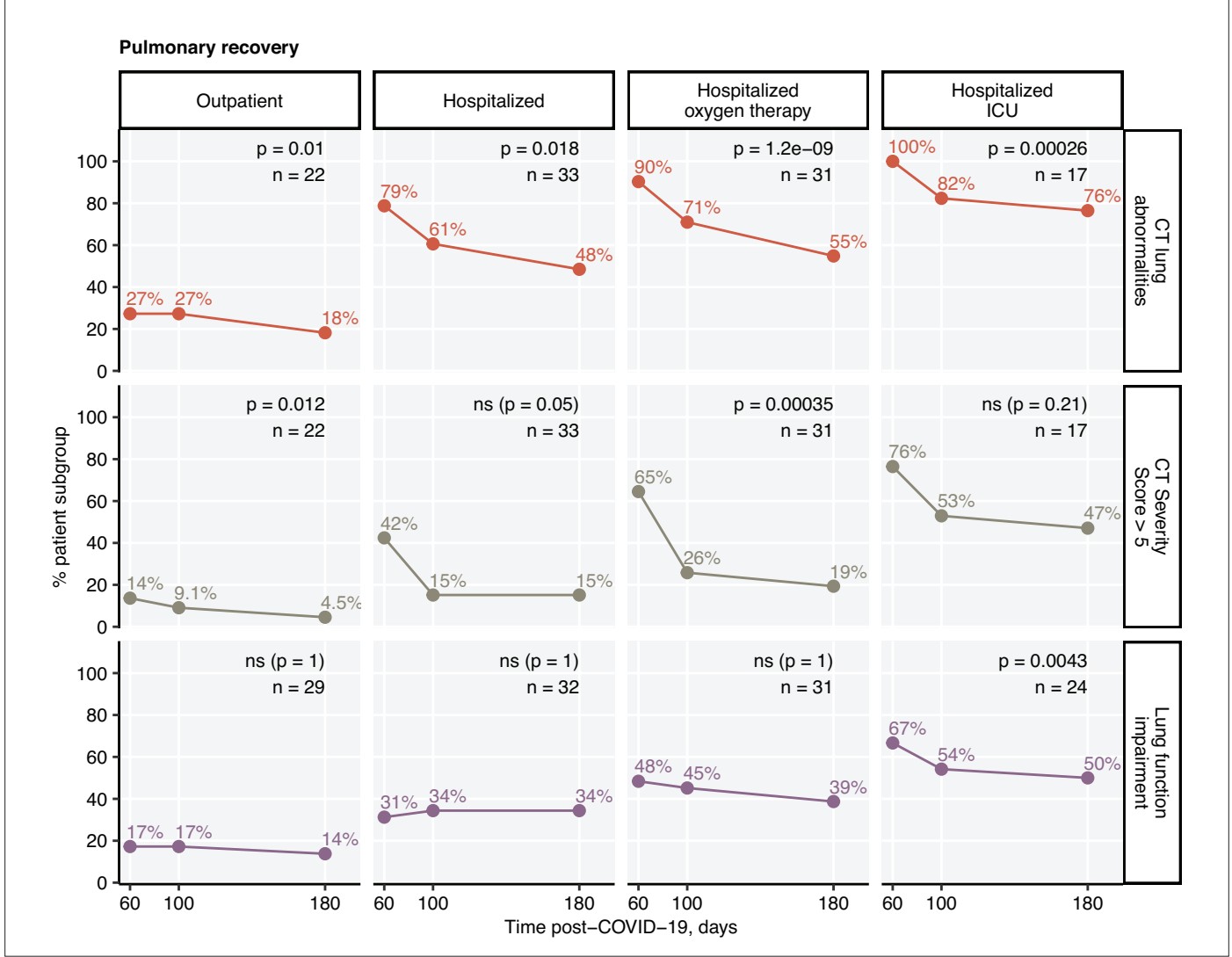

**Figure 3.** Kinetic of pulmonary recovery. Recovery from any lung computed tomography (CT) abnormalities, moderate-to-severe lung CT abnormalities (severity score > 5), and recovery from functional lung impairment were investigated in the participants stratified by acute COVID-19 severity by mixed-effect logistic modeling (random effect: individual; fixed effect: time). Significance was determined by the likelihood ratio test corrected for multiple testing with the Benjamini–Hochberg method. Frequencies of the given abnormality at the indicated time points are presented, and p-values and the numbers of complete observations are indicated in the plots.

The online version of this article includes the following figure supplement(s) for figure 3:

**Figure supplement 1.** Co-occurrence of lung computed tomography (CT) abnormalities, functional lung impairment, and any persistent symptoms.

**Figure supplement 2.** Co-occurrence of moderate-to-severe lung computed tomography (CT) abnormalities, functional lung impairment, and any persistent symptoms.

**Figure supplement 3.** Frequency of mild and moderate-to-severe lung computed tomography (CT) abnormalities.

1) were identified (*Figure 6—figure supplement 1*, *Appendix 1—table 3*): (1) cluster 1 with male sex, hypertension, and cardiovascular and metabolic comorbidity; (2) cluster 2, including characteristics of acute COVID-19 severity and inflammatory markers; and (3) cluster 3 consisting of acute and persistent COVID-19 symptoms (*Figure 6—figure supplement 2*, *Appendix 1—table 3*).

The 6-month follow-up outcome variables were incorporated in the cluster structure using kNN prediction (*Leng et al., 2013*). Long-term symptom persistence was associated with acute and long-lasting COVID-19 symptoms in cluster 3, whereas pulmonary outcome parameters were grouped with cluster 2 features (*Figure 6A*, *Figure 6—figure supplement 2*, *Appendix 1—table 3*). Preexisting comorbidities such as malignancy, kidney, lung and gastrointestinal disease, obesity, and diabetes

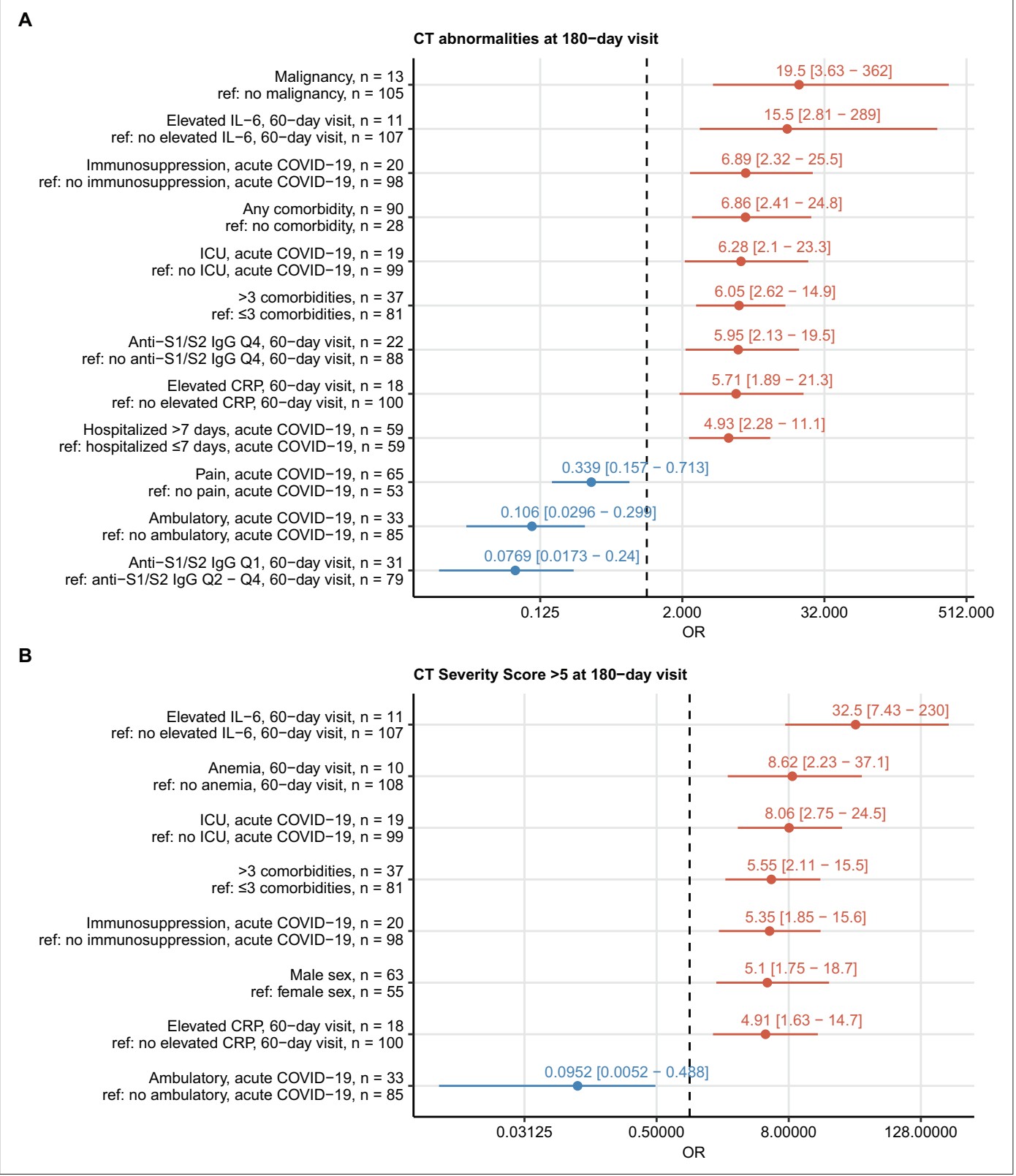

**Figure 4.** Risk factors of persistent radiological lung abnormalities. Association of 52 binary explanatory variables (*Appendix 1—table 1*) with the presence of any lung computed tomography (CT) abnormalities (**A**) or moderate-to-severe lung CT abnormalities (severity score > 5) (**B**) at the 180-day follow-up visit was investigated with a series of univariate logistic models (*Appendix 1—table 2*). Odds ratio (OR) significance was determined by Wald Z test and corrected for multiple testing with the Benjamini–Hochberg method. ORs with 95% confidence intervals for significant favorable and

*Figure 4 continued on next page*

Figure 4 continued

unfavorable factors are presented in forest plots. Model baseline (ref) and numbers of complete observations are presented in the plot axis text. Q1, Q2, Q3, Q4: first, second, third, and fourth quartile of anti-S1/S2 IgG titer; ICU: intensive care unit.

were found the closest cluster neighbors of mild CT abnormalities (severity score ≤ 5). Moderate-to-severe structural alterations (severity score > 5) and LF deficits were, in turn, tightly linked to markers of protracted systemic inflammation (IL-6, CRP, anemia of inflammation) (*Sonnweber et al., 2020*; *Figure 6B*).

## Risk stratification for perturbed pulmonary recovery by unsupervised clustering

Next, we tested whether subsets of patients at risk of an incomplete 6-month recovery may be defined by a similar clustering procedure employing exclusively non-CT and non-LF clinical variables (*Appendix 1—table 1*). Applying a combined SOM – hierarchical clustering approach, three clusters of the study participants were identified (*Figure 7*, *Figure 7—figure supplement 1*; *Vesanto and Alhoniemi, 2000*; *Kohonen, 1995*). Prolonged hospitalization, anti-infective therapy, overweight or obesity, pain during acute COVID-19, and low anti-S1/S2 titers at the 60-day follow-up were found the most influential clustering features (*Figure 7—figure supplement 2*; *Breiman, 2001*). The patient subsets identified by the SOM approach differed significantly in frequency of radiological lung abnormalities and substantially, yet not significantly, in the frequency of LF impairment at the 180-day

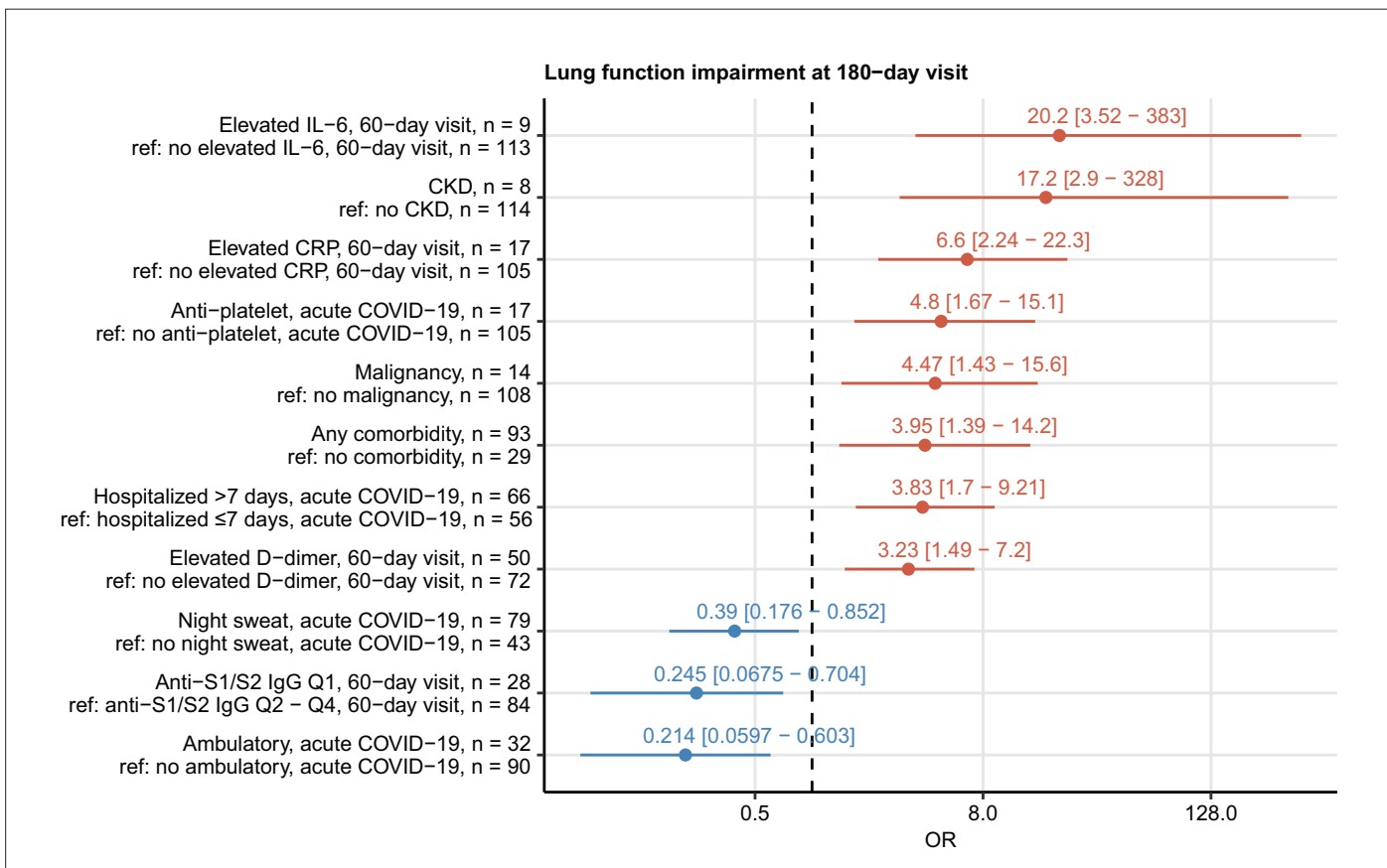

**Figure 5.** Risk factors of persistent functional lung impairment. Association of 52 binary explanatory variables (*Appendix 1—table 1*) with the presence of functional lung impairment at the 180-day follow-up visit was investigated with a series of univariate logistic models (*Appendix 1—table 2*). Odds ratio (OR) significance was determined by Wald Z test and corrected for multiple testing with the Benjamini–Hochberg method. ORs with 95% confidence intervals for the significant favorable and unfavorable factors are presented in a forest plot. Model baseline (ref) and n numbers of complete observations are presented in the plot axis text. Q1, Q2, Q3, Q4: first, second, third, and fourth quartile of anti-S1/S2 IgG titer; CKD: chronic kidney disease.

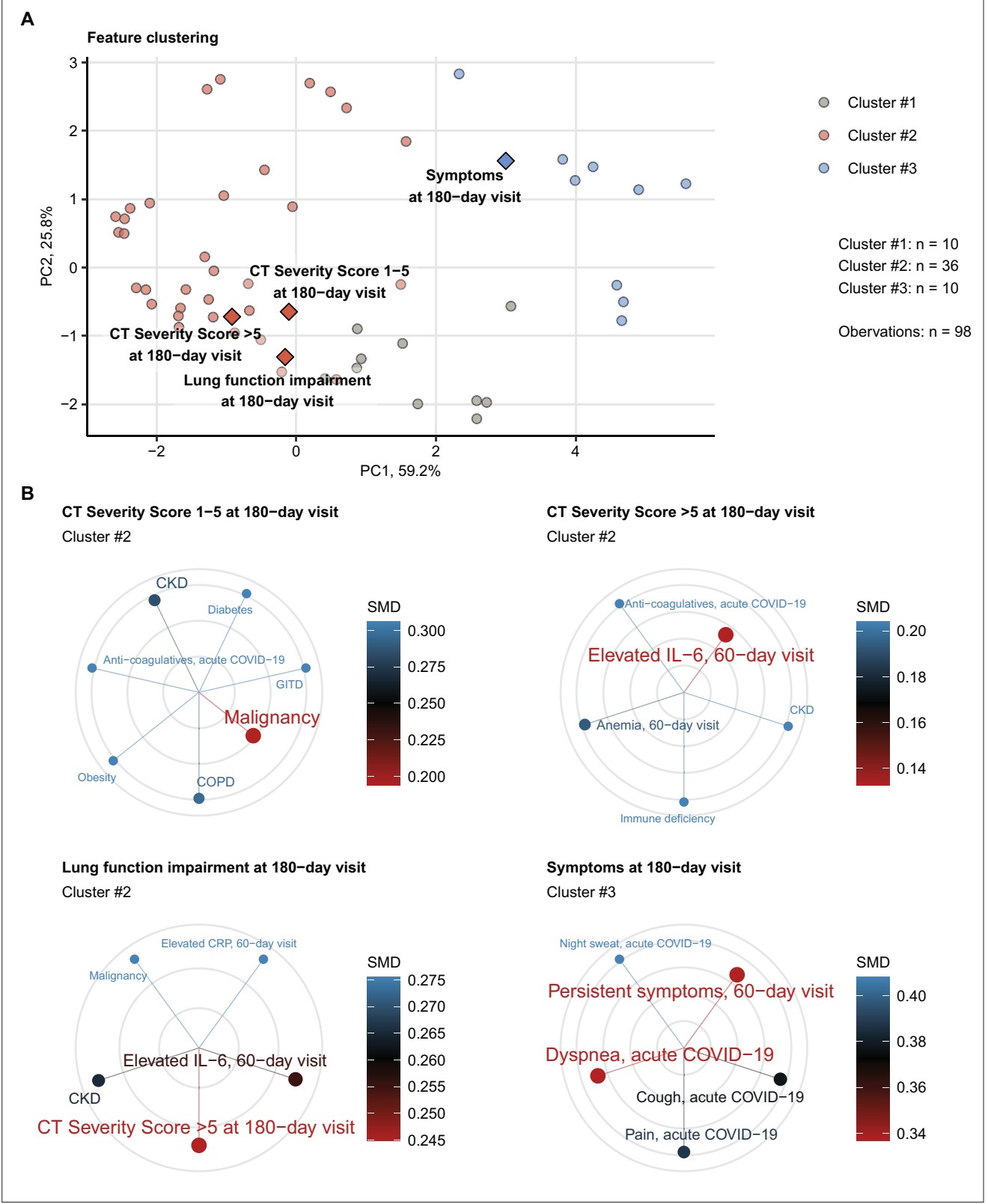

**Figure 6.** Association of incomplete symptom, lung function, and radiological lung recovery with demographic and clinical parameters of acute COVID-19 and early recovery. Clustering of 52 non-computed tomography (non-CT) and non-lung function binary explanatory variables recorded for acute COVID-19 or at the early 60-day follow-up visit (**Appendix 1—table 1**) was investigated by partitioning around medoids (PAM) algorithm with simple matching distance (SMD) dissimilarity measure (**Figure 6—figure supplement 1**, **Appendix 1—table 3**). The cluster assignment for the

*Figure 6 continued on next page*

*Figure 6 continued*

outcome variables at the 180-day follow-up visit (persistent symptoms, functional lung impairment, mild lung CT abnormalities [severity score ≤ 5] and moderate-to-severe lung CT abnormalities [severity score > 5]) was predicted by k-nearest neighbor (kNN) label propagation procedure. Numbers of complete observations and numbers of features in the clusters are indicated in (**A**). (**A**) Cluster assignment of the outcome variables (diamonds) presented in the plot of principal component (PC) scores. The first two major PCs are displayed. The explanatory variables are visualized as points. Percentages of the data set variance associated with the PC are presented in the plot axes. (**B**) Five nearest neighbors (lowest SMD) of the outcome variables presented in radial plots. Font size, point radius, and color code for SMD values. Q1, Q2, Q3, Q4: first, second, third, and fourth quartile of anti-S1/S2 IgG titer; GITD: gastrointestinal disease; CKD: chronic kidney disease; ICU: intensive care unit; COPD: chronic obstructive pulmonary disease.

The online version of this article includes the following figure supplement(s) for figure 6:

**Figure supplement 1.** Study feature clustering algorithm.

**Figure supplement 2.** Semi-supervised clustering of mild and moderate-to-severe lung computed tomography (CT) abnormalities, functional lung impairment, and persistent symptoms at the 180-day follow-up with parameters of acute COVID-19 and early convalescence.

follow-up. In particular, most of the individuals assigned to the largest, low-risk (LR) subset were CT and LF abnormality-free. The frequency and severity of radiological pulmonary findings were elevated in the smallest intermediate-risk subset (IR) and peaked in the high-risk (HR) group (*Figure 8A*). Despite a comparable frequency of long-term symptoms between the LR, IR, and HR subsets (*Figure 8A*), the HR collective showed the lowest prevalence of dyspnea, cough, night sweating, pain, gastrointestinal manifestations, and complete absence of hyposmia at the 180-day follow-up (*Figure 8B*). Although the LR subset primarily comprised mild COVID-19 cases and the HR subset ICU survivors, the cluster assignment (IR vs. LR, HR vs. LR) remained an independent correlate of persistent CT and LF abnormalities after adjustment for the acute COVID-19 severity (*Figure 8—figure supplement 1*).

## Prediction of persistent symptoms and pulmonary abnormalities by machine learning

Finally, we investigated if the 6-month follow-up outcome may be predicted by ML classifiers trained with a set of non-CT and non-LF variables recorded during acute COVID-19 and at the 60-day follow-up (*Appendix 1—table 1*). To this end, five technically unrelated ML classifiers were tested (*Appendix 1—table 4*; *Kuhn, 2008*): C5.0 (*Quinlan, 1993*), random forests (RF) (*Breiman, 2001*), support vector machines with radial kernel (SVM-R) (*Weston and Watkins, 1998*), shallow neural network (Nnet) (*Ripley, 2014*), and elastic net generalized linear regression (glmNet) (*Friedman et al., 2010*). In addition, the single classifiers with varying outcome-specific accuracy (*Figure 9—figure supplement 1*) were bundled into ensembles by the elastic net procedure (*Figure 9—figure supplement 2*, *Appendix 1—table 4*; *Kuhn, 2008*; *Deane-Mayer and Knowles, 2019*). Finally, the classifier and ensemble performance was investigated in the training cohort and 20-fold CV by ROC (*Appendix 1—table 5*).

All tested ML algorithms and ensembles demonstrated good accuracy (area under the curve [AUC] > 0.78) and sensitivity (>0.84) at predicting any lung CT abnormalities at the 6-month follow-up in the study cohort serving as a training data set. Their efficiency in CV was moderate (AUC: 0.69–0.81; sensitivity: 0.69–0.78) (*Figure 9*, *Figure 9—figure supplement 3*, *Appendix 1—table 5*). In turn, moderate-to-severe structural lung findings were recognized with markedly lower sensitivity both in the training data set (>0.43) and the CV (0.39–0.48). Even though impaired LF and persistent symptoms were common at the 6-month follow-up in the training data set (*Figures 2 and 3*), nearly half of the cases were not identified by any of the tested ML algorithms and their ensembles in the CV setting (*Figure 9*, *Figure 9—figure supplement 3*, *Appendix 1—table 5*). The sensitivity of the ensembles and single classifiers at predicting CT and LF abnormalities was substantially better in severe and critical COVID-19 survivors than in ambulatory and moderate cases (*Figure 10*, *Appendix 1—table 6*).

The most important explanatory variables for pulmonary abnormalities by three unrelated classifiers (C5.0, RF, and glmNet) included preexisting malignancy, multimorbidity, markers of systemic inflammation (IL-6 and CRP), and anti-S1/S2 antibody levels at the 60-day follow-up (*Figure 9—figure supplement 4*, *Figure 9—figure supplement 5*, *Figure 9—figure supplement 6*). The highly influential parameters at prediction of symptoms at the 180-day follow-up encompassed symptom presence at the 60-day follow-up, as well as obesity and dyspnea during acute COVID-19 (*Figure 9—figure supplement 7*).

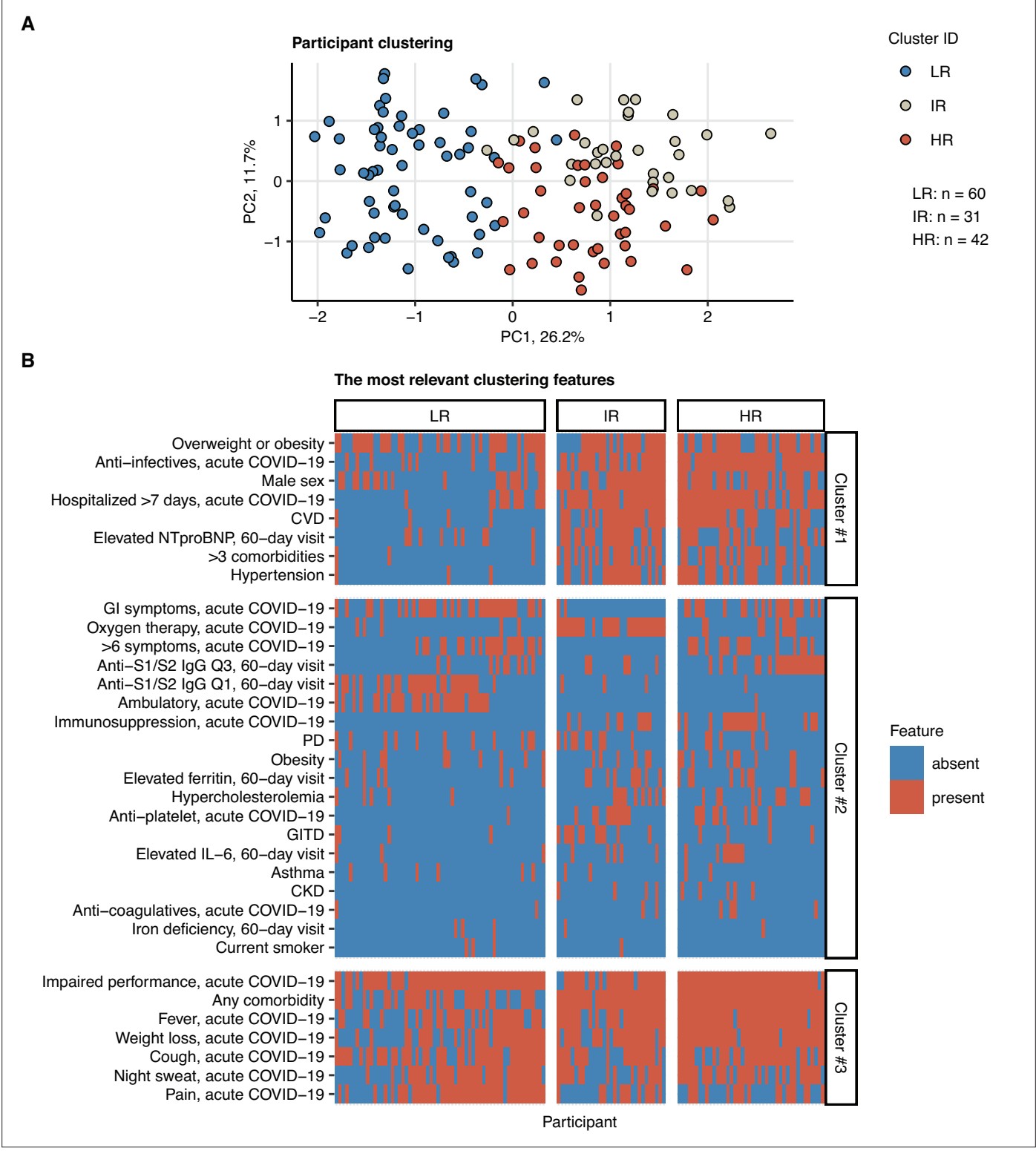

**Figure 7.** Clustering of the study participants by non-lung function and non-computed tomography (non-CT) clinical features. Study participants (n = 133 with the complete variable set) were clustered with respect to 52 non-CT and non-lung function binary explanatory variables recorded for acute COVID-19 or at the 60-day follow-up visit (**Appendix 1—table 1**) using a combined self-organizing map (SOM: simple matching distance) and hierarchical clustering (Ward.D2 method, Euclidean distance) procedure (**Figure 7—figure supplement 1**). The numbers of participants assigned to low-risk (LR), intermediate-risk (IR), and high-risk (HR) clusters are indicated in (**A**). (**A**) Cluster assignment of the study participants in the plot of principal component (PC) scores. The first two major PCs are displayed. Percentages of the data set variance associated with the PC are presented in the plot

*Figure 7 continued on next page*

Figure 7 continued

axes. (**B**) Presence of the most influential clustering features (*Figure 7—figure supplement 2*) in the participant clusters presented as a heat map. Cluster #1, #2, and #3 refer to the feature clusters defined in *Figure 6*. Q1, Q2, Q3, Q4: first, second, third, and fourth quartile of anti-S1/S2 IgG titer; GITD: gastrointestinal disease; CKD: chronic kidney disease; CVD: cardiovascular disease; GI: gastrointestinal; PD: pulmonary disease.

The online version of this article includes the following figure supplement(s) for figure 7:

**Figure supplement 1.** Study participant clustering algorithm.

**Figure supplement 2.** Impact of specific variables on the quality of participant clustering.

## Discussion

Herein, we prospectively evaluated trajectories of COVID-19 recovery in an observational cohort enrolled in the Austrian CovILD study (*Sonnweber et al., 2021*). Despite the resolution of symptoms and pulmonary abnormalities at the 6-month follow-up in a large fraction of the study participants, the recovery pace was substantially slower in the late convalescence when compared with the first three months after diagnosis (*Sonnweber et al., 2021*; *Huang et al., 2021a*). Persistent symptoms and CT findings were detected in more than 40% and reduced LF in approximately one-third of the cohort, which is in line with recovery kinetics and signs of lung lesion chronicity reported by others (*Caruso et al., 2021*; *Huang et al., 2021b*; *Huang et al., 2021a*; *Faverio et al., 2021*; *Hellemons et al., 2021*; *Zhou et al., 2021*). By comparison, similar protracted pulmonary recovery was reported for SARS (*Hui et al., 2005*; *Ng et al., 2004*; *Ngai et al., 2010*; *Lam et al., 2009*) and non-COVID-19 acute respiratory distress syndrome (*Wilcox et al., 2013*; *Masclans et al., 2011*). Of note, treatment approaches for hospitalized patients in our cohorts and similar cohorts recruited at the pandemic onset in early 2020 (*Caruso et al., 2021*; *Huang et al., 2021b*; *Huang et al., 2021a*; *Faverio et al., 2021*; *Hellemons et al., 2021*) differ significantly from the current standard of care for acute COVID-19, which includes early systemic steroid use and antiviral and various immunomodulatory medications. How improved standardized therapy and anti-SARS-CoV-2 vaccination affect the clinical and pulmonary recovery needs to be investigated.

In roughly half of our study participants with abnormal lung CT findings, and especially in those with low-grade structural abnormalities, no overt LF impairment at follow-up was discerned. Still, even subclinical lung alterations may bear the potential for clinically relevant progression of interstitial lung disease (*Suliman et al., 2015*; *Hatabu et al., 2020*) requiring systematic CT and LF monitoring. Conversely, symptom persistence was weakly associated with incomplete functional or structural pulmonary recovery.

Since PASC are found in as many as 10% of COVID-19 patients (*Sahanic et al., 2021*; *Venkatesan, 2021*; *Sudre et al., 2021b*), robust, resource-saving tools assessing the individual risk of pulmonary complications are urgently needed (*Shah et al., 2021*; *Raghu and Wilson, 2020*). Covariates and characteristics of severe acute COVID-19 such as male sex, age, and preexisting comorbidities, hospitalization, ventilation, and ICU stay were proposed as the risk factors of persistent pulmonary impairment (*Sonnweber et al., 2021*; *Caruso et al., 2021*; *Huang et al., 2021a*; *Faverio et al., 2021*; *Raghu and Wilson, 2020*). However, their applicability in predicting complications of pulmonary recovery from mild or moderate COVID-19 is limited. Our results of univariate modeling, clustering, and ML prediction point towards a distinct long-term pulmonary risk phenotype that manifests during acute COVID-19 and early recovery and whose central components are protracted systemic (IL-6, CRP, anemia of inflammation) and microvascular inflammation (D-dimer), and strong humoral response (anti-S1/S2 IgG) demographic risk factors and comorbidities (*Sonnweber et al., 2020*). Hence, consecutive monitoring of systemic inflammatory parameters analogous to concepts of interstitial lung disease in autoimmune disorders (*Khanna et al., 2020*) and anti-S1/S2 antibody levels may improve identification of the individuals at risk of chronic pulmonary damage irrespective of the acute COVID-19 severity.

Clustering and ML have been employed for deep phenotyping and predicting acute and post-acute COVID-19 outcomes in multivariable data sets (*Sahanic et al., 2021*; *Sudre et al., 2021a*; *Estiri et al., 2021*; *Demichev et al., 2021*; *Benito-León et al., 2021*). We demonstrate that subsets of COVID-19 patients that significantly differ in the risk for long-term CT abnormalities may be defined by an easily accessible clinical parameter set available at the early post-COVID-19 assessment. This approach did not involve any CT or LF variables. Furthermore, the cluster classification correlated with

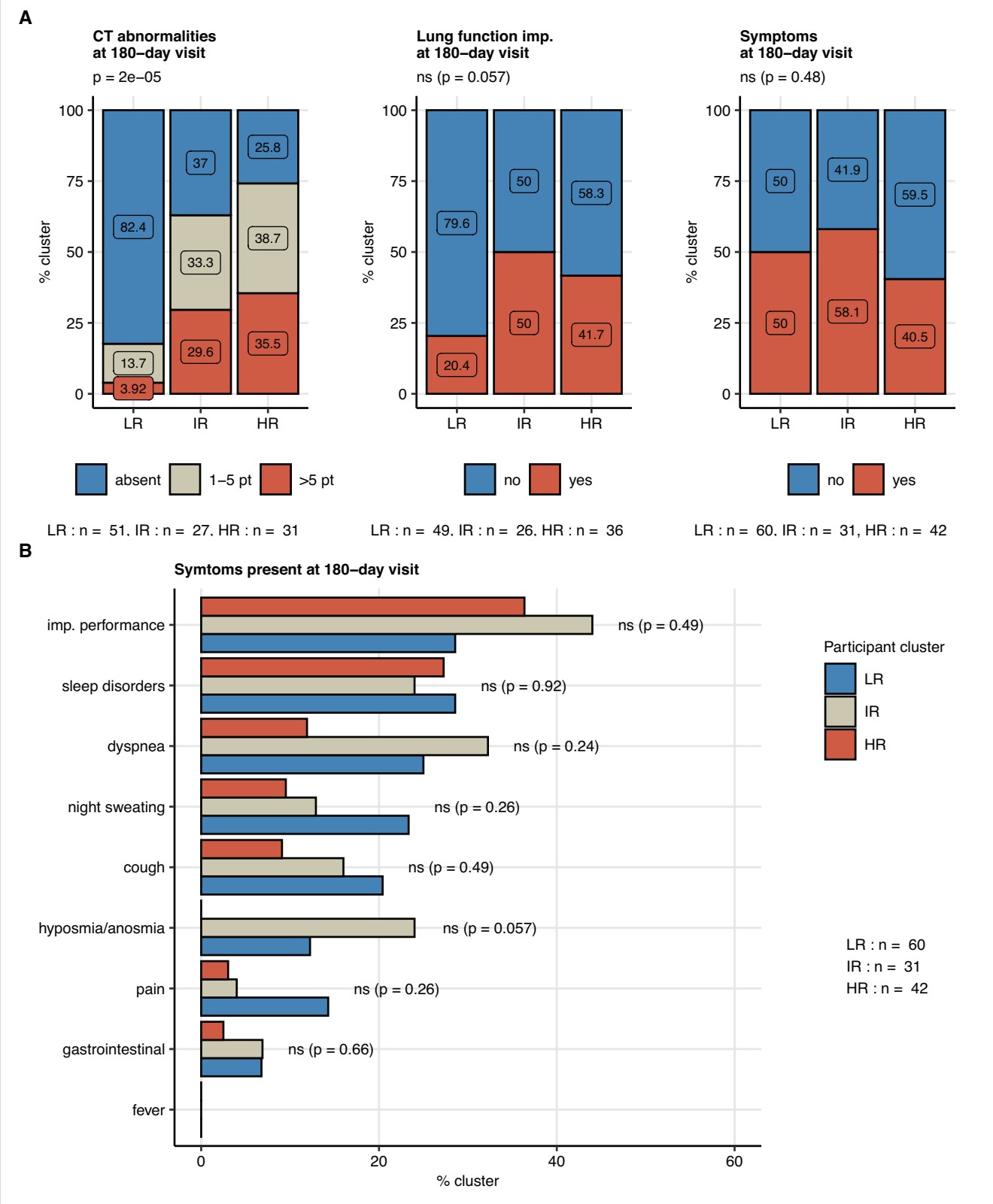

**Figure 8.** Frequency of persistent radiological lung abnormalities, functional lung impairment, and symptoms in the participant clusters. The clusters of study participants were defined by non-lung function and non-computed tomography (non-CT) features as presented in *Figure 7*. Frequencies of outcome variables at the 180-day follow-up visit (mild [severity score ≤ 5], moderate-to-severe lung CT abnormalities [severity score > 5], functional lung impairment, and persistent symptoms) were compared between the low-risk (LR), intermediate-risk (IR), and high-risk (HR) participant clusters by $\chi^2$ test

*Figure 8 continued on next page*

*Figure 8 continued*

corrected for multiple testing with the Benjamini–Hochberg method. p-Values and numbers of participants assigned to the clusters are indicated in the plots. (**A**) Frequencies of the outcome features in the participant clusters. (**B**) Frequencies of specific symptoms in the participant clusters.

The online version of this article includes the following figure supplement(s) for figure 8:

**Figure supplement 1.** Risk of radiological lung abnormalities at the 180-day follow-up in the participant clusters.

the risk of long-term pulmonary abnormalities independently of the acute COVID-19 severity. Thus, these characteristics provide a useful tool for broad screening of convalescent populations, including individuals who experienced mild or moderate COVID-19.

We show that technically unrelated ML classifiers and their ensemble trained without CT and LF explanatory variables can predict lung CT findings independently of their grading at the 6-month follow-up with good specificity and sensitivity in the training collective and CV. By contrast, the more specific prediction of moderate-to-severe lung CT or risk estimation for LF deficits demonstrated a limited sensitivity. For the moderate-to-severe CT abnormalities, this can be primarily traced back to their low frequency resulting in a suboptimal classifier training, especially in CV. A substantial fraction of the participants (20.7%, n = 30) suffered from a preexisting respiratory condition (pulmonary disease, asthma, or COPD) likely paralleled by LF reduction, which possibly confounded the prediction of the post-COVID-19 LF deficits both by clustering and ML. Accumulating evidence suggests that post-acute COVID-19 symptoms are highly heterogeneous conditions with multiorgan, neurocognitive, and psychological manifestations (*Sahanic et al., 2021*; *Evans et al., 2021*; *Davis et al., 2021*), which may differ in risk factor constellations. This could explain why univariate modeling, clustering, and ML failed to estimate persistent symptom risk in our small study cohort. In general, the ML prediction quality may greatly benefit from a larger training data set and inclusion of additional explanatory variables such as cellular readouts of inflammation, in-depth medication, and broader acute symptom data. Nevertheless, the herein described cluster- and ML classifiers represent resource-effective tools that may assist in the screening of medical record data and identification of COVID-19 patients requiring systematic CT and LF monitoring. To facilitate the identification of patients at risk for protracted respiratory recovery and enable validation in an external collective, we implemented the clustering and prediction procedures in an open-source risk assessment application (https://im2-ibk.shinyapps.io/CovILD/).

Our study bears limitations primarily concerning the low sample size and the cross-sectional character of the trial. Because of the impaired availability of the patients and the prolonged inpatient rehabilitation, the 60- and 100-day follow-up visits in part showed a temporal overlap that may have impacted the accuracy of the longitudinal data. Missingness of the consecutive outcome variable record and the participant dropout, particularly of mild and moderate COVID-19 cases, may have also potentially confounded the participant clustering results and ML risk estimation for CT abnormalities and LF impairment since prolonged hospitalization was found to be a crucial cluster-defining and influential explanatory feature. Additionally, even though the reproducibility of the risk assessment algorithms was partially addressed by CV, cluster and ML classifiers call for verification in a larger, independent multicenter collective of COVID-19 convalescents.

In summary, in our CovILD study cohort we found a high frequency of CT and LF abnormalities and persistent symptoms at the 6-month follow-up, and a flattened recovery kinetics after 3 months post-COVID-19. Systematic risk modeling reveled a set of clinical variables linked to protracted pulmonary recovery apart from the severity of acute infection such as inflammatory markers, anti-S1/S2 IgG levels, multimorbidity, and male sex. We demonstrate that clustering and ML classifiers may help to identify individuals at risk of persistent lung lesions and to relocate medical resources to prevent long-term disability.

## Acknowledgements

We acknowledge the commitment of the staff and providers of our institutions through the COVID-19 crisis and the suffering and loss of our patients as well as their families. PT is (from May 2021 on) a freelance data scientist working in his own enterprise 'Data Analytics as a Service Tirol'. He received an

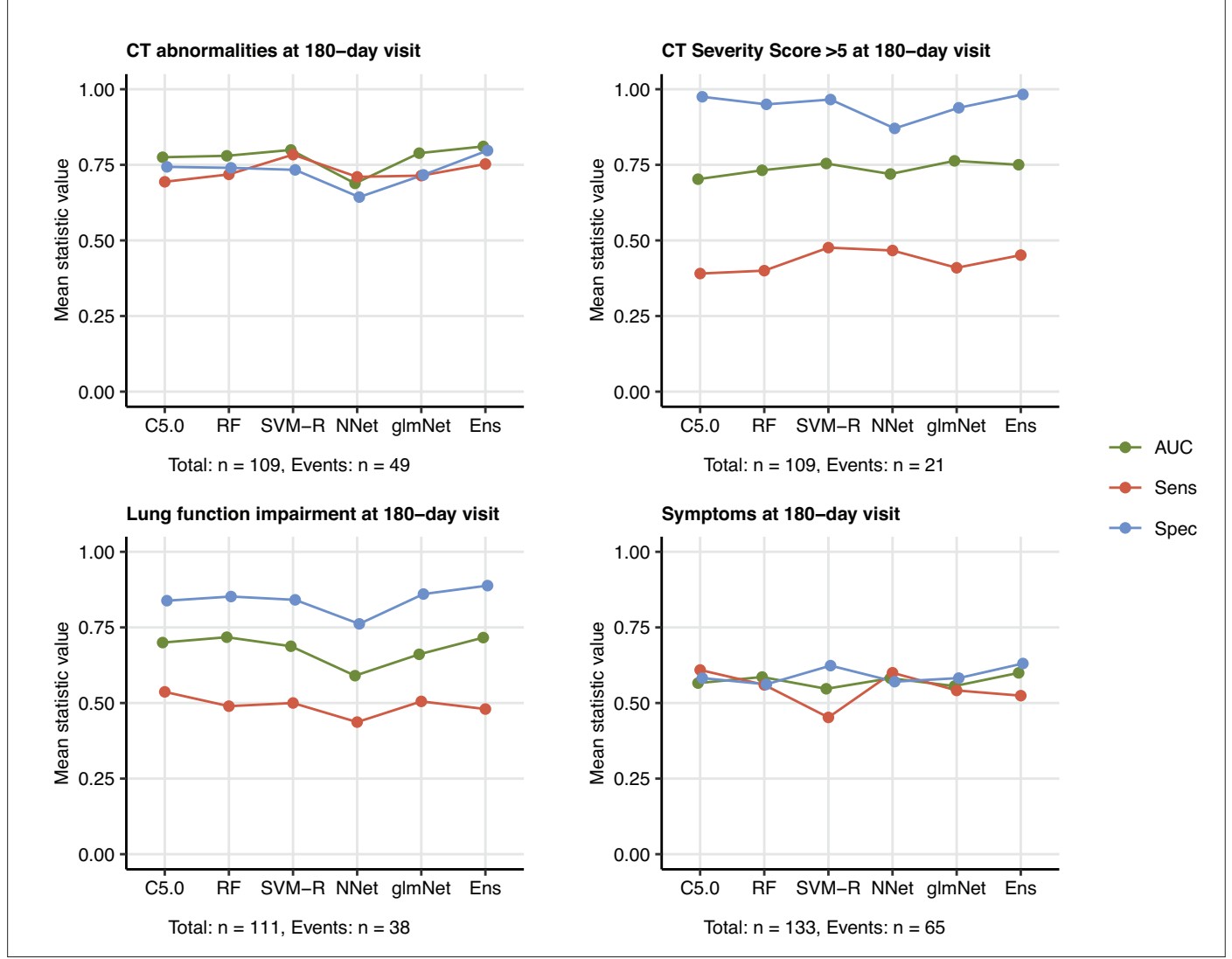

**Figure 9.** Prediction of persistent radiological lung abnormalities, functional lung impairment, and symptoms by machine learning algorithms. Single machine learning classifiers (C5.0; RF: random forests; SVM-R: support vector machines with radial kernel; NNet: neural network; glmNet: elastic net) and their ensemble (Ens) were trained in the cohort data set with 52 non-computed tomography (non-CT) and non-lung function binary explanatory variables recorded for acute COVID-19 or at the 60-day follow-up visit (*Appendix 1—table 1*) for predicting outcome variables at the 180-day follow-up visit (any lung CT abnormalities, moderate-to-severe lung CT abnormalities [severity score > 5], functional lung impairment, and persistent symptoms) (*Appendix 1—table 4*). The prediction accuracy was verified by repeated 20-fold cross-validation (five repeats). Receiver-operating characteristics (ROCs) of the algorithms in the cross-validation are presented: area under the curve (AUC), sensitivity (Sens), and specificity (Spec) (*Appendix 1—table 5*). The numbers of complete observations and outcome events are indicated under the plots.

The online version of this article includes the following figure supplement(s) for figure 9:

**Figure supplement 1.** Correlation of the machine learning algorithm prediction accuracy.

**Figure supplement 2.** Machine learning model ensembles.

**Figure supplement 3.** Prediction of persistent radiological lung abnormalities, functional lung impairment, and symptoms by machine learning algorithms in the training data sets.

**Figure supplement 4.** Variable importance statistics for prediction of lung computed tomography (CT) abnormalities at the 180-day follow-up by machine learning classifiers.

**Figure supplement 5.** Variable importance statistics for prediction of moderate-to-severe lung computed tomography (CT) abnormalities at the 180-day follow-up by machine learning classifiers.

**Figure supplement 6.** Variable importance statistics for prediction of functional lung impairment at the 180-day follow-up by machine learning classifiers.

**Figure supplement 7.** Variable importance statistics for prediction of persistent symptoms at the 180-day follow-up by machine learning classifiers.

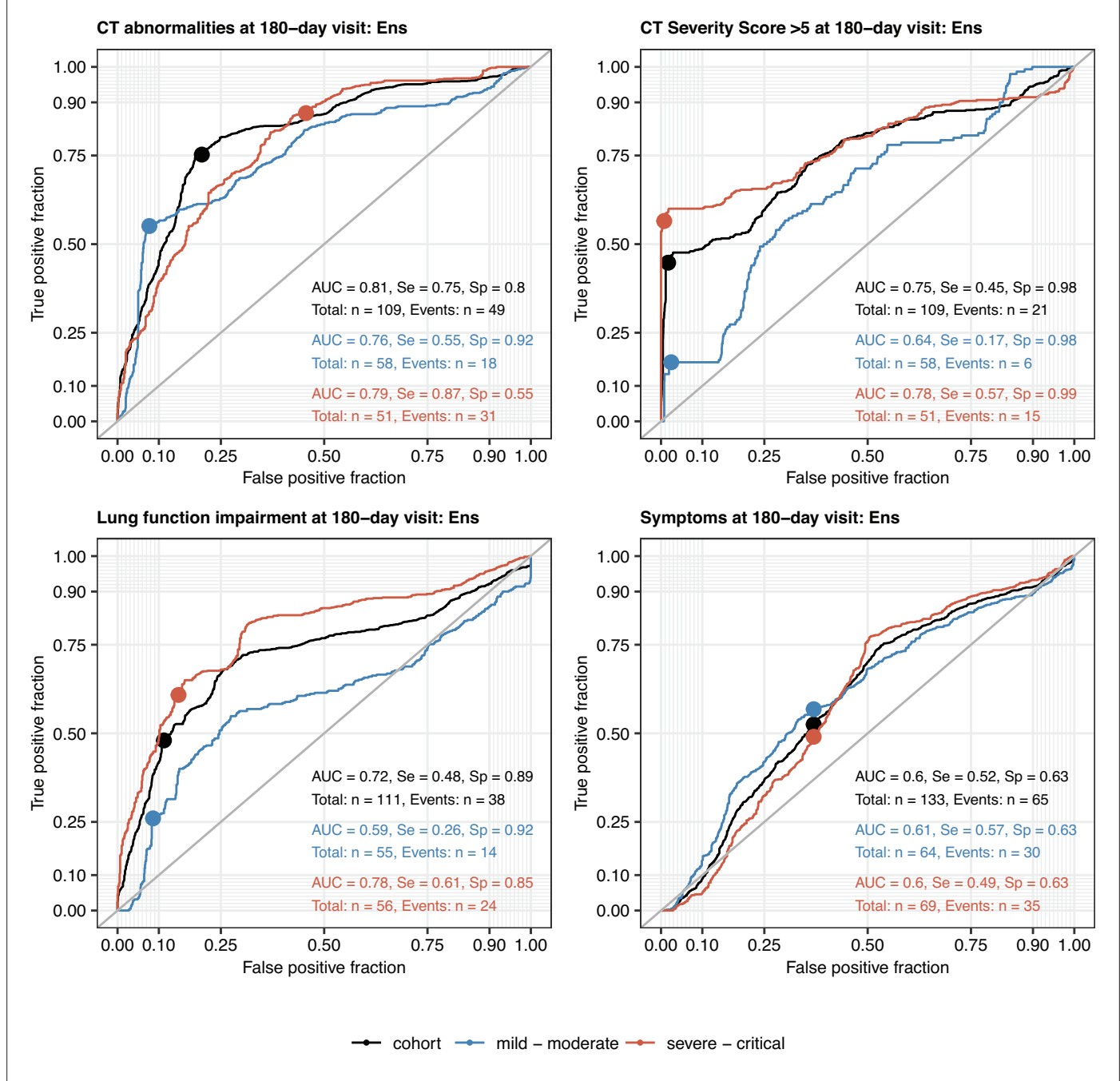

**Figure 10.** Performance of the machine learning ensemble classifier in mild-to-moderate and severe-to-critical COVID-19 convalescents. The machine learning classifier ensemble (Ens) was developed as presented in *Figure 9*. Its performance at predicting outcome variables at the 180-day follow-up visit (any computed tomography [CT] lung abnormalities, moderate-to-severe lung CT abnormalities [severity score > 5], functional lung impairment, and persistent symptoms) in the entire cohort, mild-to-moderate (outpatient or hospitalized without oxygen), and severe-to-critical COVID-19 convalescents (oxygen therapy or ICU) in repeated 20-fold cross-validation (five repeats) was assessed by receiver-operating characteristic (ROC) (*Appendix 1—table 6*). ROC curves and statistics (AUC: area under the curve; Se: sensitivity; Sp: specificity) in the cross-validation are shown. Numbers of complete observations and outcome events are indicated in the plots.

honorary for the study data management, curation and analysis and minor manuscript work. The other authors declare no conflict of interest related to this study. The study was funded by the research fund of the state of Tyrol (Project GZ 71934, JLR) and an Investigator-Initiated Study grant by Boehringer Ingelheim (IIS 1199-0424, IT). The funding bodies did not influence the development of the research and manuscript.

## Additional information

### Competing interests

Piotr Tymoszuk: owns his own business, Data Analytics as a Service Tirol, for which he performs freelance data science work. Has also received an honorarium for the study data management, curation and analysis and minor manuscript work. The author has no other competing interests to declare. The other authors declare that no competing interests exist.

### Funding

| Funder | Grant reference number | Author |
|---|---|---|
| Land Tirol | GZ 71934 | Judith Löffler-Ragg |
| Boehringer Ingelheim | IIS 1199-0424 | Ivan Tancevski |

The funders had no role in study design, data collection and interpretation, or the decision to submit the work for publication.

### Author contributions

Thomas Sonnweber, Conceptualization, Data curation, Formal analysis, Investigation, Methodology, Project administration, Resources, Supervision, Validation, Visualization, Writing – original draft, Writing – review and editing; Piotr Tymoszuk, Data curation, Formal analysis, Investigation, Methodology, Project administration, Software, Writing – original draft, Writing – review and editing; Sabina Sahanic, Gerlig Widmann, Conceptualization, Investigation, Methodology, Project administration, Resources; Anna Boehm, Alex Pizzini, Conceptualization, Investigation, Methodology, Project administration; Anna Luger, Katharina Kurz, Sabine Koppelstätter, Magdalena Aichner, Bernhard Puchner, Alexander Egger, Gregor Hoermann, Ewald Wöll, Investigation, Methodology, Project administration; Christoph Schwabl, Manfred Nairz, Investigation, Methodology, Project administration, Resources; Philipp Grubwieser, Methodology, Resources; Günter Weiss, Conceptualization, Investigation, Methodology, Project administration, Resources, Writing – review and editing; Ivan Tancevski, Conceptualization, Data curation, Funding acquisition, Investigation, Methodology, Project administration, Resources, Supervision, Writing – original draft, Writing – review and editing; Judith Löffler-Ragg, Conceptualization, Data curation, Formal analysis, Funding acquisition, Investigation, Methodology, Project administration, Supervision, Validation, Writing – original draft, Writing – review and editing

### Author ORCIDs

Thomas Sonnweber http://orcid.org/0000-0002-5080-386X
Piotr Tymoszuk http://orcid.org/0000-0002-0398-6034
Anna Luger http://orcid.org/0000-0002-0445-8372
Ivan Tancevski http://orcid.org/0000-0001-5116-8960
Judith Löffler-Ragg http://orcid.org/0000-0003-0873-7501

### Ethics

Human subjects: All participants gave written informed consent. The study was approved by the institutional review board at the Medical University of Innsbruck (approval number: 1103/2020), and registered at ClinicalTrials.gov (NCT04416100).

### Decision letter and Author response

Decision letter https://doi.org/10.7554/eLife.72500.sa1
Author response https://doi.org/10.7554/eLife.72500.sa2

## Additional files

### Supplementary files
- Transparent reporting form
- Source data 1. Figure plots source data.

### Data availability
The complete R analysis pipeline and the anonymized study data in form of stratified study variables are available as a public GitHub repository: https://github.com/PiotrTymoszuk/CovILD_6_Months (copy archived at swh:1:rev:df521ede1d284e074a0484d3e4d0ce71097d00c3). The R code for the key tools used for uni-variate modeling and model quality control (Figures 4 and 5, https://github.com/PiotrTymoszuk/lmqc; copy archived at swh:1:rev:a020119d8f23b60901115c5c2ce6f6c71998ed31), cluster analysis and its quality control (Figures 6–7, https://github.com/PiotrTymoszuk/clustering-tools-2; copy archived at swh:1:rev:64141197ca28838a8978dce9093443537157d79f) and the risk assessment applicaiton (https://github.com/PiotrTymoszuk/COVILD-recovery-assessment-app; copy archived at swh:1:rev:95f02215f4c13425d3b76f6a13b7862a53279ab9) is available at GitHub. Source data for Figures 2–10 has been included as Source data 1.

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

# Appendix 1

**Appendix 1—table 1.** Study variables.

Variable: variable name in the analysis pipeline; reference time point: study visit, the variable was recorded at; label: variable label in figures and tables.

| Variable | Reference time point | Label | Variable type | Stratification cutoff |
|---|---|---|---|---|
| sex_male_V0 | Acute COVID-19 | Male sex | Explanatory | |
| obesity_rec_V0 | Acute COVID-19 | Obesity | Explanatory | BMI > 30 kg/m$^2$ |
| current_smoker_V0 | Acute COVID-19 | Current smoker | Explanatory | |
| smoking_ex_V0 | Acute COVID-19 | Ex-smoker | Explanatory | |
| CVDis_rec_V0 | Acute COVID-19 | CVD | Explanatory | |
| hypertension_rec_V0 | Acute COVID-19 | Hypertension | Explanatory | |
| PDis_rec_V0 | Acute COVID-19 | PD | Explanatory | |
| COPD_rec_V0 | Acute COVID-19 | COPD | Explanatory | |
| asthma_rec_V0 | Acute COVID-19 | Asthma | Explanatory | |
| endocrine_metabolic_rec_V0 | Acute COVID-19 | Metabolic disorders | Explanatory | |
| hypercholesterolemia_rec_V0 | Acute COVID-19 | Hypercholesterolemia | Explanatory | |
| diabetes_rec_V0 | Acute COVID-19 | Diabetes | Explanatory | |
| CKDis_rec_V0 | Acute COVID-19 | CKD | Explanatory | |
| GITDis_rec_V0 | Acute COVID-19 | GITD | Explanatory | |
| malignancy_rec_V0 | Acute COVID-19 | Malignancy | Explanatory | |
| immune_deficiency_rec_V0 | Acute COVID-19 | Immune deficiency | Explanatory | |
| weight_change_rec_V0 | Acute COVID-19 | Weight loss, acute COVID-19 | Explanatory | ≥1 kg |
| dyspnoe_rec_V0 | Acute COVID-19 | Dyspnea, acute COVID-19 | Explanatory | |
| cough_rec_V0 | Acute COVID-19 | Cough, acute COVID-19 | Explanatory | |
| fever_rec_V0 | Acute COVID-19 | Fever, acute COVID-19 | Explanatory | |
| night_sweat_rec_V0 | Acute COVID-19 | Night sweat, acute COVID-19 | Explanatory | |
| pain_rec_V0 | Acute COVID-19 | Pain, acute COVID-19 | Explanatory | |
| GI_sympt_rec_V0 | Acute COVID-19 | GI symptoms, acute COVID-19 | Explanatory | |
| anosmia_rec_V0 | Acute COVID-19 | Anosmia, acute COVID-19 | Explanatory | |
| ECOG_imp_rec_V0 | Acute COVID-19 | Impaired performance, acute COVID-19 | Explanatory | ECOG ≥ 1 |
| sleep_disorder_rec_V0 | Acute COVID-19 | Sleep disorders, acute COVID-19 | Explanatory | |
| treat_antiinfec_rec_V0 | Acute COVID-19 | Anti-infectives, acute COVID-19 | Explanatory | |
| treat_antiplat_rec_V0 | Acute COVID-19 | Antiplatelet, acute COVID-19 | Explanatory | |
| treat_anticoag_rec_V0 | Acute COVID-19 | Anticoagulatives, acute COVID-19 | Explanatory | |
| treat_immunosuppr_rec_V0 | Acute COVID-19 | Immunosuppression, acute COVID-19 | Explanatory | |
| anemia_rec_V1 | 60-day follow-up | Anemia, 60-day visit | Explanatory | Male: Hb < 14 g/dL; female: Hb <12 g/dL |
| ferr_elv_rec_V1 | 60-day follow-up | Elevated ferritin, 60-day visit | Explanatory | Male: > 300 ng/mL; female: > 150 ng/mL |

*Appendix 1—table 1 Continued on next page*

*Appendix 1—table 1 Continued*

| Variable | Reference time point | Label | Variable type | Stratification cutoff |
|---|---|---|---|---|
| NTelv_rec_V1 | 60-day follow-up | Elevated NTproBNP, 60-day visit | Explanatory | >125 pg/mL |
| Ddimerelv_rec_V1 | 60-day follow-up | Elevated D-dimer, 60-day visit | Explanatory | >500 pg/mL FEU |
| CRP_elv_rec_V1 | 60-day follow-up | Elevated CRP, 60-day visit | Explanatory | >0.5 mg/dL |
| IL6_elv_rec_V1 | 60-day follow-up | Elevated IL-6, 60-day visit | Explanatory | >7 pg/mL |
| iron_deficiency_30_rec_V1 | 60-day follow-up | Iron deficiency, 60-day visit | Explanatory | TF-saturation < 15% |
| age_65_V0 | Acute COVID-19 | Age over 65 | Explanatory | >65 years |
| hosp_7d_V0 | Acute COVID-19 | Hospitalized > 7 days, acute COVID-19 | Explanatory | >7 days |
| comorb_present_V0 | Acute COVID-19 | Any comorbidity | Explanatory | >0 comorbidities |
| comorb_3_V0 | Acute COVID-19 | >3 comorbidities | Explanatory | >3 comorbidities |
| overweight_V0 | Acute COVID-19 | Overweight or obesity | Explanatory | BMI > 25 kg/m$^2$ |
| sympt_6_V0 | Acute COVID-19 | >6 symptoms, acute COVID-19 | Explanatory | >6 symptoms |
| sympt_present_V1 | 60-day follow-up | Persistent symptoms, 60-day visit | Explanatory | >0 symptoms at 180-day visit |
| ab_0_V1 | 60-day follow-up | Anti-S1/S2 IgG Q1, 60-day visit | Explanatory | (0, 312] BAU/mL |
| ab_25_V1 | 60-day follow-up | Anti-S1/S2 IgG Q2, 60-day visit | Explanatory | (312, 644] BAU/mL |
| ab_50_V1 | 60-day follow-up | Anti-S1/S2 IgG Q3, 60-day visit | Explanatory | (644, 975] BAU/mL |
| ab_75_V1 | 60-day follow-up | Anti-S1/S2 IgG Q4, 60-day visit | Explanatory | > 975 BAU/mL |
| pat_group_G1_V0 | Acute COVID-19 | Ambulatory, acute COVID-19 | Explanatory | |
| pat_group_G2_V0 | Acute COVID-19 | Hospitalized, acute COVID-19 | Explanatory | |
| pat_group_G3_V0 | Acute COVID-19 | Oxygen therapy, acute COVID-19 | Explanatory | |
| pat_group_G4_V0 | Acute COVID-19 | ICU, acute COVID-19 | Explanatory | |
| CT_findings_V3 | 180-day follow-up | CT abnormalities at 180-day visit | Outcome | |
| CT_sev_low_V3 | 180-day follow-up | CT severity score 1–5 at 180-day visit | Outcome | |
| CTsevabove5_V3 | 180-day follow-up | CT severity score >5 at 180-day visit | Outcome | |
| sympt_present_V3 | 180-day follow-up | Symptoms at 180-day visit | Outcome | |
| lung_function_impaired_V3 | 180-day follow-up | Lung function impairment at 180-day visit | Outcome | |

CVD = cardiovascular disease; PD = pulmonary disease; COPD = chronic obstructive pulmonary disease; CKD = chronic kidney disease; GITD = gastrointestinal disease; GI = gastrointestinal; CRP = C-reactive protein; ICU = intensive care unit; CT = computed tomography; BMI = body mass index; BAU = binding antibody unit.

**Appendix 1—table 2.** Results of univariate risk modeling.

Outcome: outcome variable at the 180-day follow-up visit; covariate: explanatory variable; baseline: reference level of the explanatory variable; OR: odds ratios with 95% confidence intervals; pFDR: significanct p-value corrected for multiple testing with the Benjamini–Hochberg method (FDR: false discovery rate).

| Outcome | Covariate | Baseline | Complete cases | OR | pFDR |
|---|---|---|---|---|---|
| CT abnormalities at 180-day visit | Male sex, n = 63 | No male sex, n = 55 | 118 | 3.79 [1.77–8.44] | p=0.01 |
| CT abnormalities at 180-day visit | Obesity, n = 22 | No obesity, n = 96 | 118 | 1.07 [0.415–2.72] | ns (p=0.9) |
| CT abnormalities at 180-day visit | Current smoker, n = 4 | No current smoker, n = 114 | 118 | 0.412 [0.02–3.33] | ns (p=0.51) |
| CT abnormalities at 180-day visit | Ex-smoker, n = 48 | No ex-smoker, n = 70 | 118 | 1.5 [0.716–3.16] | ns (p=0.36) |
| CT abnormalities at 180-day visit | CVD, n = 45 | No CVD, n = 73 | 118 | 3.36 [1.57–7.43] | p=0.012 |
| CT abnormalities at 180-day visit | Hypertension, n = 34 | No hypertension, n = 84 | 118 | 3.97 [1.73–9.54] | p=0.01 |
| CT abnormalities at 180-day visit | PD, n = 24 | No PD, n = 94 | 118 | 2.06 [0.837–5.25] | ns (p=0.2) |
| CT abnormalities at 180-day visit | COPD, n = 6 | No COPD, n = 112 | 118 | 2.67 [0.499–19.8] | ns (p=0.34) |
| CT abnormalities at 180-day visit | Asthma, n = 9 | No asthma, n = 109 | 118 | 1.02 [0.24–4.04] | ns (p=0.99) |
| CT abnormalities at 180-day visit | Metabolic disorders, n = 50 | No metabolic disorders, n = 68 | 118 | 3.14 [1.48–6.81] | p=0.017 |
| CT abnormalities at 180-day visit | Hypercholesterolemia, n = 22 | No hypercholesterolemia, n = 96 | 118 | 2.67 [1.04–7.27] | ns (p=0.093) |
| CT abnormalities at 180-day visit | Diabetes, n = 18 | No diabetes, n = 100 | 118 | 4.07 [1.41–13.5] | p=0.041 |
| CT abnormalities at 180-day visit | GITD, n = 17 | No GITD, n = 101 | 118 | 3.66 [1.25–12.2] | ns (p=0.061) |
| CT abnormalities at 180-day visit | Malignancy, n = 13 | No malignancy, n = 105 | 118 | 19.5 [3.63–362] | p=0.021 |
| CT abnormalities at 180-day visit | Immune deficiency, n = 5 | No immune deficiency, n = 113 | 118 | 1.96 [0.313–15.3] | ns (p=0.53) |
| CT abnormalities at 180-day visit | Weight loss, acute COVID-19, n = 84 | No weight loss, acute COVID-19, n = 34 | 118 | 4.45 [1.83–12.1] | p=0.011 |
| CT abnormalities at 180-day visit | Dyspnea, acute COVID-19, n = 81 | No dyspnea, acute COVID-19, n = 37 | 118 | 1.45 [0.661–3.27] | ns (p=0.43) |
| CT abnormalities at 180-day visit | Cough, acute COVID-19, n = 83 | No cough, acute COVID-19, n = 35 | 118 | 1.07 [0.484–2.41] | ns (p=0.89) |
| CT abnormalities at 180-day visit | Fever, acute COVID-19, n = 83 | No fever, acute COVID-19, n = 35 | 118 | 2.56 [1.12–6.21] | ns (p=0.072) |
| CT abnormalities at 180-day visit | Night sweat, acute COVID-19, n = 74 | No night sweat, acute COVID-19, n = 44 | 118 | 1.93 [0.902–4.26] | ns (p=0.17) |
| CT abnormalities at 180-day visit | Pain, acute COVID-19, n = 65 | No pain, acute COVID-19, n = 53 | 118 | 0.339 [0.157–0.713] | p=0.021 |
| CT abnormalities at 180-day visit | GI symptoms, acute COVID-19, n = 47 | No GI symptoms, acute COVID-19, n = 71 | 118 | 0.675 [0.316–1.42] | ns (p=0.38) |
| CT abnormalities at 180-day visit | Anosmia, acute COVID-19, n = 53 | No anosmia, acute COVID-19, n = 65 | 118 | 1.09 [0.526–2.28] | ns (p=0.85) |
| CT abnormalities at 180-day visit | Impaired performance, acute COVID-19, n = 106 | No impaired performance, acute COVID-19, n = 12 | 118 | 1.12 [0.335–3.98] | ns (p=0.89) |
| CT abnormalities at 180-day visit | Sleep disorders, acute COVID-19, n = 40 | No sleep disorders, acute COVID-19, n = 77 | 117 | 0.887 [0.407–1.91] | ns (p=0.82) |
| CT abnormalities at 180-day visit | Anti-infectives, acute COVID-19, n = 64 | No anti-infectives, acute COVID-19, n = 54 | 118 | 3.56 [1.67–7.9] | p=0.01 |
| CT abnormalities at 180-day visit | Antiplatelet, acute COVID-19, n = 12 | No antiplatelet, acute COVID-19, n = 106 | 118 | 4.4 [1.23–20.7] | ns (p=0.077) |

| Outcome | Covariate | Baseline | Complete cases | OR | pFDR |
|---|---|---|---|---|---|
| CT abnormalities at 180-day visit | Anticoagulatives, acute COVID-19, n = 4 | No anticoagulatives, acute COVID-19, n = 114 | 118 | 3.98 [0.493–81.8] | ns (p=0.32) |
| CT abnormalities at 180-day visit | Immunosuppression, acute COVID-19, n = 20 | No immunosuppression, acute COVID-19, n = 98 | 118 | 6.89 [2.32–25.5] | p=0.01 |
| CT abnormalities at 180-day visit | Anemia, 60-day visit, n = 10 | No anemia, 60-day visit, n = 108 | 118 | 5.82 [1.38–39.8] | ns (p=0.072) |
| CT abnormalities at 180-day visit | Elevated ferritin, 60-day visit, n = 20 | No elevated ferritin, 60-day visit, n = 98 | 118 | 2.18 [0.825–6.01] | ns (p=0.2) |
| CT abnormalities at 180-day visit | Elevated NTproBNP, 60-day visit, n = 38 | No elevated NTproBNP, 60-day visit, n = 80 | 118 | 2.29 [1.05–5.1] | ns (p=0.084) |
| CT abnormalities at 180-day visit | Elevated D-dimer, 60-day visit, n = 49 | No elevated D-dimer, 60-day visit, n = 69 | 118 | 2.9 [1.37–6.28] | p=0.023 |
| CT abnormalities at 180-day visit | Elevated CRP, 60-day visit, n = 18 | No elevated CRP, 60-day visit, n = 100 | 118 | 5.71 [1.89–21.3] | p=0.019 |
| CT abnormalities at 180-day visit | Elevated IL-6, 60-day visit, n = 11 | No elevated IL-6, 60-day visit, n = 107 | 118 | 15.5 [2.81–289] | p=0.036 |
| CT abnormalities at 180-day visit | Iron deficiency, 60-day visit, n = 6 | No iron deficiency, 60-day visit, n = 112 | 118 | 0.239 [0.0123–1.55] | ns (p=0.29) |
| CT abnormalities at 180-day visit | Age over 65, n = 32 | No age over 65, n = 86 | 118 | 2.81 [1.23–6.66] | p=0.045 |
| CT abnormalities at 180-day visit | Hospitalized >7 days, acute COVID-19, n = 59 | No hospitalized >7 days, acute COVID-19, n = 59 | 118 | 4.93 [2.28–11.1] | p=0.0026 |
| CT abnormalities at 180-day visit | Any comorbidity, n = 90 | No any comorbidity, n = 28 | 118 | 6.86 [2.41–24.8] | p=0.01 |
| CT abnormalities at 180-day visit | >3 comorbidities, n = 37 | No >3 comorbidities, n = 81 | 118 | 6.05 [2.62–14.9] | p=0.0026 |
| CT abnormalities at 180-day visit | Overweight or obesity, n = 72 | No overweight or obesity, n = 46 | 118 | 1.61 [0.762–3.48] | ns (p=0.3) |
| CT abnormalities at 180-day visit | >6 symptoms, acute COVID-19, n = 33 | No >6 symptoms, acute COVID-19, n = 85 | 118 | 0.767 [0.333–1.73] | ns (p=0.59) |
| CT abnormalities at 180-day visit | Persistent symptoms, 60-day visit, n = 93 | No persistent symptoms, 60-day visit, n = 25 | 118 | 1.91 [0.769–5.08] | ns (p=0.26) |
| CT abnormalities at 180-day visit | Anti-S1/S2 IgG Q1, 60-day visit, n = 31 | No anti-S1/S2 IgG Q1, 60-day visit, n = 79 | 110 | 0.0769 [0.0173–0.24] | p=0.0026 |
| CT abnormalities at 180-day visit | Anti-S1/S2 IgG Q2, 60-day visit, n = 30 | No anti-S1/S2 IgG Q2, 60-day visit, n = 80 | 110 | 1.12 [0.481–2.62] | ns (p=0.83) |
| CT abnormalities at 180-day visit | Anti-S1/S2 IgG Q3, 60-day visit, n = 27 | No anti-S1/S2 IgG Q3, 60-day visit, n = 83 | 110 | 1.8 [0.753–4.4] | ns (p=0.28) |
| CT abnormalities at 180-day visit | Anti-S1/S2 IgG Q4, 60-day visit, n = 22 | No anti-S1/S2 IgG Q4, 60-day visit, n = 88 | 110 | 5.95 [2.13–19.5] | p=0.01 |
| CT abnormalities at 180-day visit | Ambulatory, acute COVID-19, n = 33 | No ambulatory, acute COVID-19, n = 85 | 118 | 0.106 [0.0296–0.299] | p=0.0026 |
| CT abnormalities at 180-day visit | Hospitalized, acute COVID-19, n = 33 | No hospitalized, acute COVID-19, n = 85 | 118 | 1.28 [0.569–2.88] | ns (p=0.61) |
| CT abnormalities at 180-day visit | Oxygen therapy, acute COVID-19, n = 33 | No oxygen therapy, acute COVID-19, n = 85 | 118 | 1.52 [0.676–3.43] | ns (p=0.38) |
| CT abnormalities at 180-day visit | ICU, acute COVID-19, n = 19 | No ICU, acute COVID-19, n = 99 | 118 | 6.28 [2.1–23.3] | p=0.012 |
| CT severity score >5 at 180-day visit | Male sex, n = 63 | No male sex, n = 55 | 118 | 5.1 [1.75–18.7] | p=0.01 |
| CT severity score >5 at 180-day visit | Obesity, n = 22 | No obesity, n = 96 | 118 | 0.38 [0.0577–1.46] | ns (p=0.26) |

| Outcome | Covariate | Baseline | Complete cases | OR | pFDR |
|---|---|---|---|---|---|
| CT severity score >5 at 180-day visit | Current smoker, n = 4 | No current smoker, n = 114 | 118 | 1.48 [0.0711–12.2] | ns (p=0.77) |
| CT severity score >5 at 180-day visit | Ex-smoker, n = 48 | No ex-smoker, n = 70 | 118 | 1.59 [0.623–4.09] | ns (p=0.37) |
| CT severity score >5 at 180-day visit | CVD, n = 45 | No CVD, n = 73 | 118 | 4.71 [1.8–13.5] | p=0.0042 |
| CT severity score >5 at 180-day visit | Hypertension, n = 34 | No hypertension, n = 84 | 118 | 3.17 [1.21–8.38] | p=0.029 |
| CT severity score >5 at 180-day visit | PD, n = 24 | No PD, n = 94 | 118 | 2.17 [0.735–6.02] | ns (p=0.18) |
| CT severity score >5 at 180-day visit | COPD, n = 6 | No COPD, n = 112 | 118 | 2.3 [0.304–12.7] | ns (p=0.39) |
| CT severity score >5 at 180-day visit | Asthma, n = 9 | No asthma, n = 109 | 118 | 2.37 [0.468–9.85] | ns (p=0.29) |
| CT severity score >5 at 180-day visit | Metabolic disorders, n = 50 | No metabolic disorders, n = 68 | 118 | 2.92 [1.14–7.95] | p=0.045 |
| CT severity score >5 at 180-day visit | Hypercholesterolemia, n = 22 | No hypercholesterolemia, n = 96 | 118 | 2.52 [0.845–7.12] | ns (p=0.12) |
| CT severity score >5 at 180-day visit | Diabetes, n = 18 | No diabetes, n = 100 | 118 | 2.63 [0.816–7.87] | ns (p=0.12) |
| CT severity score >5 at 180-day visit | CKD, n = 6 | No CKD, n = 112 | 118 | 4.89 [0.851–28.3] | ns (p=0.091) |
| CT severity score >5 at 180-day visit | GITD, n = 17 | No GITD, n = 101 | 118 | 2.9 [0.892–8.83] | ns (p=0.092) |
| CT severity score >5 at 180-day visit | Malignancy, n = 13 | No malignancy, n = 105 | 118 | 0.333 [0.0178–1.84] | ns (p=0.35) |
| CT severity score >5 at 180-day visit | Immune deficiency, n = 5 | No immune deficiency, n = 113 | 118 | 7.42 [1.16–59.3] | ns (p=0.052) |
| CT severity score >5 at 180-day visit | Weight loss, acute COVID-19, n = 84 | No weight loss, acute COVID-19, n = 34 | 118 | 3.02 [0.939–13.5] | ns (p=0.13) |
| CT severity score >5 at 180-day visit | Dyspnea, acute COVID-19, n = 81 | No dyspnea, acute COVID-19, n = 37 | 118 | 1.7 [0.609–5.54] | ns (p=0.38) |
| CT severity score >5 at 180-day visit | Cough, acute COVID-19, n = 83 | No cough, acute COVID-19, n = 35 | 118 | 0.537 [0.206–1.44] | ns (p=0.25) |
| CT severity score >5 at 180-day visit | Fever, acute COVID-19, n = 83 | No fever, acute COVID-19, n = 35 | 118 | 2.15 [0.727–7.9] | ns (p=0.24) |
| CT severity score >5 at 180-day visit | Night sweat, acute COVID-19, n = 74 | No night sweat, acute COVID-19, n = 44 | 118 | 2.33 [0.84–7.55] | ns (p=0.17) |
| CT severity score >5 at 180-day visit | Pain, acute COVID-19, n = 65 | No pain, acute COVID-19, n = 53 | 118 | 0.495 [0.187–1.26] | ns (p=0.18) |

| Outcome | Covariate | Baseline | Complete cases | OR | pFDR |
|---|---|---|---|---|---|
| CT severity score >5 at 180-day visit | GI symptoms, acute COVID-19, n = 47 | No GI symptoms, acute COVID-19, n = 71 | 118 | 0.503 [0.168–1.34] | ns (p=0.23) |
| CT severity score >5 at 180-day visit | Anosmia, acute COVID-19, n = 53 | No anosmia, acute COVID-19, n = 65 | 118 | 1.61 [0.634–4.16] | ns (p=0.36) |
| CT severity score >5 at 180-day visit | Impaired performance, acute COVID-19, n = 106 | No impaired performance, acute COVID-19, n = 12 | 118 | 2.72 [0.486–51] | ns (p=0.39) |
| CT severity score >5 at 180-day visit | Sleep disorders, acute COVID-19, n = 40 | No sleep disorders, acute COVID-19, n = 77 | 117 | 1.13 [0.412–2.91] | ns (p=0.84) |
| CT severity score >5 at 180-day visit | Anti-infectives, acute COVID-19, n = 64 | No anti-infectives, acute COVID-19, n = 54 | 118 | 4.89 [1.68–17.9] | p=0.012 |
| CT severity score >5 at 180-day visit | Antiplatelet, acute COVID-19, n = 12 | No antiplatelet, acute COVID-19, n = 106 | 118 | 3.74 [1.01–13.2] | ns (p=0.06) |
| CT severity score >5 at 180-day visit | Anticoagulatives, acute COVID-19, n = 4 | No anticoagulatives, acute COVID-19, n = 114 | 118 | 4.7 [0.538–41.1] | ns (p=0.17) |
| CT severity score >5 at 180-day visit | Immunosuppression, acute COVID-19, n = 20 | No immunosuppression, acute COVID-19, n = 98 | 118 | 5.35 [1.85–15.6] | p=0.0036 |
| CT severity score >5 at 180-day visit | Anemia, 60-day visit, n = 10 | No anemia, 60-day visit, n = 108 | 118 | 8.62 [2.23–37.1] | p=0.0039 |
| CT severity score >5 at 180-day visit | Elevated ferritin, 60-day visit, n = 20 | No elevated ferritin, 60-day visit, n = 98 | 118 | 2.2 [0.693–6.42] | ns (p=0.2) |
| CT severity score >5 at 180-day visit | Elevated NTproBNP, 60-day visit, n = 38 | No elevated NTproBNP, 60-day visit, n = 80 | 118 | 3.23 [1.25–8.55] | p=0.026 |
| CT severity score >5 at 180-day visit | Elevated D-dimer, 60-day visit, n = 49 | No elevated D-dimer, 60-day visit, n = 69 | 118 | 2.41 [0.945–6.38] | ns (p=0.096) |
| CT severity score >5 at 180-day visit | Elevated CRP, 60-day visit, n = 18 | No elevated CRP, 60-day visit, n = 100 | 118 | 4.91 [1.63–14.7] | p=0.0075 |
| CT severity score >5 at 180-day visit | Elevated IL-6, 60-day visit, n = 11 | No elevated IL-6, 60-day visit, n = 107 | 118 | 32.5 [7.43–230] | p=7.5e-05 |
| CT severity score >5 at 180-day visit | Iron deficiency, 60-day visit, n = 6 | No iron deficiency, 60-day visit, n = 112 | 118 | 0.867 [0.044–5.75] | ns (p=0.92) |
| CT severity score >5 at 180-day visit | Age over 65, n = 32 | No age over 65, n = 86 | 118 | 2.8 [1.05–7.4] | ns (p=0.055) |
| CT severity score >5 at 180-day visit | Hospitalized >7 days, acute COVID-19, n = 59 | No hospitalized >7 days, acute COVID-19, n = 59 | 118 | 4.37 [1.58–14.2] | p=0.012 |
| CT severity score >5 at 180-day visit | Any comorbidity, n = 90 | No any comorbidity, n = 28 | 118 | 8.22 [1.59–151] | ns (p=0.065) |
| CT severity score >5 at 180-day visit | >3 comorbidities, n = 37 | No >3 comorbidities, n = 81 | 118 | 5.55 [2.11–15.5] | p=0.0013 |
| CT severity score >5 at 180-day visit | Overweight or obesity, n = 72 | No overweight or obesity, n = 46 | 118 | 0.72 [0.282–1.87] | ns (p=0.53) |

| Outcome | Covariate | Baseline | Complete cases | OR | pFDR |
|---------|-----------|----------|----------------|-----|------|
| CT severity score >5 at 180-day visit | >6 symptoms, acute COVID-19, n = 33 | No >6 symptoms, acute COVID-19, n = 85 | 118 | 1.26 [0.438–3.35] | ns (p=0.69) |
| CT severity score >5 at 180-day visit | Persistent symptoms, 60-day visit, n = 93 | No persistent symptoms, 60-day visit, n = 25 | 118 | 3.15 [0.831–20.7] | ns (p=0.18) |
| CT severity score >5 at 180-day visit | Anti-S1/S2 IgG Q2, 60-day visit, n = 30 | No anti-S1/S2 IgG Q2, 60-day visit, n = 80 | 110 | 1.87 [0.666–5.07] | ns (p=0.26) |
| CT severity score >5 at 180-day visit | Anti-S1/S2 IgG Q3, 60-day visit, n = 27 | No anti-S1/S2 IgG Q3, 60-day visit, n = 83 | 110 | 0.675 [0.18–2.05] | ns (p=0.55) |
| CT severity score >5 at 180-day visit | Anti-S1/S2 IgG Q4, 60-day visit, n = 22 | No anti-S1/S2 IgG Q4, 60-day visit, n = 88 | 110 | 4.38 [1.53–12.6] | p=0.01 |
| CT severity score >5 at 180-day visit | Ambulatory, acute COVID-19, n = 33 | No ambulatory, acute COVID-19, n = 85 | 118 | 0.0952 [0.0052–0.488] | p=0.039 |
| CT severity score >5 at 180-day visit | Hospitalized, acute COVID-19, n = 33 | No hospitalized, acute COVID-19, n = 85 | 118 | 0.714 [0.218–2.01] | ns (p=0.58) |
| CT severity score >5 at 180-day visit | Oxygen therapy, acute COVID-19, n = 33 | No oxygen therapy, acute COVID-19, n = 85 | 118 | 0.958 [0.316–2.61] | ns (p=0.95) |
| CT severity score >5 at 180-day visit | ICU, acute COVID-19, n = 19 | No ICU, acute COVID-19, n = 99 | 118 | 8.06 [2.75–24.5] | p=0.00035 |
| Symptoms at 180-day visit | Male sex, n = 82 | No male sex, n = 63 | 145 | 0.701 [0.361–1.35] | ns (p=0.97) |
| Symptoms at 180-day visit | Obesity, n = 28 | No obesity, n = 117 | 145 | 0.42 [0.169–0.982] | ns (p=0.84) |
| Symptoms at 180-day visit | Current smoker, n = 4 | No current smoker, n = 141 | 145 | 3.22 [0.401–66] | ns (p=0.97) |
| Symptoms at 180-day visit | Ex-smoker, n = 57 | No ex-smoker, n = 88 | 145 | 1.27 [0.654–2.49] | ns (p=0.97) |
| Symptoms at 180-day visit | CVD, n = 58 | No CVD, n = 87 | 145 | 0.851 [0.436–1.66] | ns (p=0.97) |
| Symptoms at 180-day visit | Hypertension, n = 44 | No hypertension, n = 101 | 145 | 0.931 [0.456–1.89] | ns (p=0.97) |
| Symptoms at 180-day visit | PD, n = 27 | No PD, n = 118 | 145 | 1.38 [0.598–3.26] | ns (p=0.97) |
| Symptoms at 180-day visit | COPD, n = 8 | No COPD, n = 137 | 145 | 1.04 [0.238–4.58] | ns (p=0.97) |
| Symptoms at 180-day visit | Asthma, n = 10 | No asthma, n = 135 | 145 | 1.05 [0.279–3.92] | ns (p=0.97) |
| Symptoms at 180-day visit | Metabolic disorders, n = 63 | No metabolic disorders, n = 82 | 145 | 1.02 [0.527–1.96] | ns (p=0.97) |
| Symptoms at 180-day visit | Hypercholesterolemia, n = 27 | No hypercholesterolemia, n = 118 | 145 | 0.55 [0.226–1.28] | ns (p=0.97) |
| Symptoms at 180-day visit | Diabetes, n = 24 | No diabetes, n = 121 | 145 | 1.05 [0.434–2.54] | ns (p=0.97) |
| Symptoms at 180-day visit | CKD, n = 10 | No CKD, n = 135 | 145 | 1.62 [0.442–6.56] | ns (p=0.97) |
| Symptoms at 180-day visit | GITD, n = 20 | No GITD, n = 125 | 145 | 1.68 [0.649–4.55] | ns (p=0.97) |
| Symptoms at 180-day visit | Malignancy, n = 17 | No malignancy, n = 128 | 145 | 0.7 [0.241–1.94] | ns (p=0.97) |

| Outcome | Covariate | Baseline | Complete cases | OR | pFDR |
|---|---|---|---|---|---|
| Symptoms at 180-day visit | Immune deficiency, n = 9 | No immune deficiency, n = 136 | 145 | 0.824 [0.197–3.24] | ns (p=0.97) |
| Symptoms at 180-day visit | Weight loss, acute COVID-19, n = 106 | No weight loss, acute COVID-19, n = 39 | 145 | 1.34 [0.644–2.84] | ns (p=0.97) |
| Symptoms at 180-day visit | Dyspnea, acute COVID-19, n = 98 | No dyspnea, acute COVID-19, n = 47 | 145 | 2.84 [1.39–6.04] | ns (p=0.2) |
| Symptoms at 180-day visit | Cough, acute COVID-19, n = 102 | No cough, acute COVID-19, n = 43 | 145 | 1.97 [0.96–4.17] | ns (p=0.88) |
| Symptoms at 180-day visit | Fever, acute COVID-19, n = 106 | No fever, acute COVID-19, n = 39 | 145 | 1.17 [0.559–2.45] | ns (p=0.97) |
| Symptoms at 180-day visit | Night sweat, acute COVID-19, n = 92 | No night sweat, acute COVID-19, n = 53 | 145 | 1.42 [0.723–2.83] | ns (p=0.97) |
| Symptoms at 180-day visit | Pain, acute COVID-19, n = 78 | No pain, acute COVID-19, n = 67 | 145 | 1.92 [0.993–3.75] | ns (p=0.84) |
| Symptoms at 180-day visit | GI symptoms, acute COVID-19, n = 59 | No GI symptoms, acute COVID-19, n = 86 | 145 | 1.27 [0.656–2.48] | ns (p=0.97) |
| Symptoms at 180-day visit | Anosmia, acute COVID-19, n = 62 | No anosmia, acute COVID-19, n = 83 | 145 | 1.69 [0.874–3.31] | ns (p=0.96) |
| Symptoms at 180-day visit | Impaired performance, acute COVID-19, n = 132 | No impaired performance, acute COVID-19, n = 13 | 145 | 1.13 [0.358–3.69] | ns (p=0.97) |
| Symptoms at 180-day visit | Sleep disorders, acute COVID-19, n = 56 | No sleep disorders, acute COVID-19, n = 88 | 144 | 1.38 [0.708–2.73] | ns (p=0.97) |
| Symptoms at 180-day visit | Anti-infectives, acute COVID-19, n = 78 | No anti-infectives, acute COVID-19, n = 67 | 145 | 0.701 [0.362–1.35] | ns (p=0.97) |
| Symptoms at 180-day visit | Antiplatelet, acute COVID-19, n = 22 | No antiplatelet, acute COVID-19, n = 123 | 145 | 1.05 [0.42–2.63] | ns (p=0.97) |
| Symptoms at 180-day visit | Anticoagulatives, acute COVID-19, n = 9 | No anticoagulatives, acute COVID-19, n = 136 | 145 | 2.18 [0.553–10.7] | ns (p=0.97) |
| Symptoms at 180-day visit | Immunosuppression, acute COVID-19, n = 27 | No immunosuppression, acute COVID-19, n = 118 | 145 | 1.38 [0.598–3.26] | ns (p=0.97) |
| Symptoms at 180-day visit | Anemia, 60-day visit, n = 16 | No anemia, 60-day visit, n = 129 | 145 | 0.591 [0.191–1.69] | ns (p=0.97) |
| Symptoms at 180-day visit | Elevated ferritin, 60-day visit, n = 26 | No elevated ferritin, 60-day visit, n = 118 | 144 | 1.29 [0.551–3.07] | ns (p=0.97) |
| Symptoms at 180-day visit | Elevated NTproBNP, 60-day visit, n = 52 | No elevated NTproBNP, 60-day visit, n = 93 | 145 | 1.96 [0.987–3.94] | ns (p=0.84) |
| Symptoms at 180-day visit | Elevated D-dimer, 60-day visit, n = 60 | No elevated D-dimer, 60-day visit, n = 85 | 145 | 1.7 [0.874–3.33] | ns (p=0.96) |
| Symptoms at 180-day visit | Elevated CRP, 60-day visit, n = 23 | No elevated CRP, 60-day visit, n = 122 | 145 | 1.16 [0.475–2.88] | ns (p=0.97) |
| Symptoms at 180-day visit | Elevated IL-6, 60-day visit, n = 17 | No elevated IL-6, 60-day visit, n = 128 | 145 | 0.529 [0.173–1.48] | ns (p=0.97) |
| Symptoms at 180-day visit | Iron deficiency, 60-day visit, n = 6 | No iron deficiency, 60-day visit, n = 138 | 144 | 2.18 [0.412–16.1] | ns (p=0.97) |
| Symptoms at 180-day visit | Age over 65, n = 43 | No age over 65, n = 102 | 145 | 1.69 [0.827–3.51] | ns (p=0.97) |
| Symptoms at 180-day visit | Hospitalized >7 days, acute COVID-19, n = 80 | No hospitalized >7 days, acute COVID-19, n = 65 | 145 | 1.1 [0.569–2.12] | ns (p=0.97) |
| Symptoms at 180-day visit | Any comorbidity, n = 112 | No any comorbidity, n = 33 | 145 | 1.03 [0.47–2.24] | ns (p=0.97) |
| Symptoms at 180-day visit | >3 comorbidities, n = 47 | No >3 comorbidities, n = 98 | 145 | 1.46 [0.727–2.95] | ns (p=0.97) |
| Symptoms at 180-day visit | Overweight or obesity, n = 86 | No overweight or obesity, n = 59 | 145 | 0.7 [0.358–1.36] | ns (p=0.97) |

| Outcome | Covariate | Baseline | Complete cases | OR | pFDR |
|---|---|---|---|---|---|
| Symptoms at 180-day visit | >6 symptoms, acute COVID-19, n = 42 | No >6 symptoms, acute COVID-19, n = 103 | 145 | 1.82 [0.885–3.82] | ns (p=0.96) |
| Symptoms at 180-day visit | Persistent symptoms, 60-day visit, n = 115 | No persistent symptoms, 60-day visit, n = 30 | 145 | 4.12 [1.71–11.1] | ns (p=0.2) |
| Symptoms at 180-day visit | Anti-S1/S2 IgG Q1, 60-day visit, n = 34 | No anti-S1/S2 IgG Q1, 60-day visit, n = 100 | 134 | 1.04 [0.476–2.28] | ns (p=0.97) |
| Symptoms at 180-day visit | Anti-S1/S2 IgG Q2, 60-day visit, n = 33 | No anti-S1/S2 IgG Q2, 60-day visit, n = 101 | 134 | 1.13 [0.512–2.49] | ns (p=0.97) |
| Symptoms at 180-day visit | Anti-S1/S2 IgG Q3, 60-day visit, n = 34 | No anti-S1/S2 IgG Q3, 60-day visit, n = 100 | 134 | 0.646 [0.289–1.41] | ns (p=0.97) |
| Symptoms at 180-day visit | Anti-S1/S2 IgG Q4, 60-day visit, n = 33 | No anti-S1/S2 IgG Q4, 60-day visit, n = 101 | 134 | 1.32 [0.603–2.95] | ns (p=0.97) |
| Symptoms at 180-day visit | Ambulatory, acute COVID-19, n = 36 | No ambulatory, acute COVID-19, n = 109 | 145 | 0.911 [0.426–1.94] | ns (p=0.97) |
| Symptoms at 180-day visit | Hospitalized, acute COVID-19, n = 37 | No hospitalized, acute COVID-19, n = 108 | 145 | 0.983 [0.463–2.08] | ns (p=0.97) |
| Symptoms at 180-day visit | Oxygen therapy, acute COVID-19, n = 40 | No oxygen therapy, acute COVID-19, n = 105 | 145 | 0.922 [0.442–1.91] | ns (p=0.97) |
| Symptoms at 180-day visit | ICU, acute COVID-19, n = 32 | No ICU, acute COVID-19, n = 113 | 145 | 1.24 [0.564–2.74] | ns (p=0.97) |
| Lung function impairment at 180-day visit | Male sex, n = 71 | No male sex, n = 51 | 122 | 2.12 [0.964–4.85] | ns (p=0.1) |
| Lung function impairment at 180-day visit | Obesity, n = 22 | No obesity, n = 100 | 122 | 1.94 [0.746–5] | ns (p=0.22) |
| Lung function impairment at 180-day visit | Current smoker, n = 3 | No current smoker, n = 119 | 122 | 4.26 [0.397–93.4] | ns (p=0.3) |
| Lung function impairment at 180-day visit | Ex-smoker, n = 45 | No ex-smoker, n = 77 | 122 | 1.95 [0.897–4.26] | ns (p=0.13) |
| Lung function impairment at 180-day visit | CVD, n = 49 | No CVD, n = 73 | 122 | 1.57 [0.727–3.39] | ns (p=0.31) |
| Lung function impairment at 180-day visit | Hypertension, n = 35 | No hypertension, n = 87 | 122 | 1.56 [0.683–3.54] | ns (p=0.34) |
| Lung function impairment at 180-day visit | PD, n = 23 | No PD, n = 99 | 122 | 2.21 [0.869–5.62] | ns (p=0.13) |
| Lung function impairment at 180-day visit | COPD, n = 7 | No COPD, n = 115 | 122 | 2.93 [0.615–15.5] | ns (p=0.23) |
| Lung function impairment at 180-day visit | Asthma, n = 9 | No asthma, n = 113 | 122 | 1.03 [0.208–4.12] | ns (p=0.97) |
| Lung function impairment at 180-day visit | Metabolic disorders, n = 53 | No metabolic disorders, n = 69 | 122 | 1.73 [0.807–3.73] | ns (p=0.22) |
| Lung function impairment at 180-day visit | Hypercholesterolemia, n = 24 | No hypercholesterolemia, n = 98 | 122 | 1.03 [0.383–2.61] | ns (p=0.96) |
| Lung function impairment at 180-day visit | Diabetes, n = 21 | No diabetes, n = 101 | 122 | 2.15 [0.816–5.64] | ns (p=0.16) |

| Outcome | Covariate | Baseline | Complete cases | OR | pFDR |
|---|---|---|---|---|---|
| Lung function impairment at 180-day visit | CKD, n = 8 | No CKD, n = 114 | 122 | 17.2 [2.9–328] | p=0.02 |
| Lung function impairment at 180-day visit | GITD, n = 16 | No GITD, n = 106 | 122 | 3.11 [1.07–9.42] | ns (p=0.061) |
| Lung function impairment at 180-day visit | Malignancy, n = 14 | No malignancy, n = 108 | 122 | 4.47 [1.43–15.6] | p=0.025 |
| Lung function impairment at 180-day visit | Immune deficiency, n = 6 | No immune deficiency, n = 116 | 122 | 2.14 [0.38–12] | ns (p=0.43) |
| Lung function impairment at 180-day visit | Weight loss, acute COVID-19, n = 91 | No weight loss, acute COVID-19, n = 31 | 122 | 1.56 [0.645–4.08] | ns (p=0.41) |
| Lung function impairment at 180-day visit | Dyspnea, acute COVID-19, n = 82 | No dyspnea, acute COVID-19, n = 40 | 122 | 3.17 [1.31–8.58] | p=0.029 |
| Lung function impairment at 180-day visit | Cough, acute COVID-19, n = 88 | No cough, acute COVID-19, n = 34 | 122 | 0.856 [0.375–2.01] | ns (p=0.76) |
| Lung function impairment at 180-day visit | Fever, acute COVID-19, n = 92 | No fever, acute COVID-19, n = 30 | 122 | 1.19 [0.496–3.01] | ns (p=0.76) |
| Lung function impairment at 180-day visit | Night sweat, acute COVID-19, n = 79 | No night sweat, acute COVID-19, n = 43 | 122 | 0.39 [0.176–0.852] | p=0.033 |
| Lung function impairment at 180-day visit | Pain, acute COVID-19, n = 65 | No pain, acute COVID-19, n = 57 | 122 | 0.609 [0.282–1.3] | ns (p=0.26) |
| Lung function impairment at 180-day visit | GI symptoms, acute COVID-19, n = 46 | No GI symptoms, acute COVID-19, n = 76 | 122 | 0.715 [0.316–1.57] | ns (p=0.47) |
| Lung function impairment at 180-day visit | Anosmia, acute COVID-19, n = 51 | No anosmia, acute COVID-19, n = 71 | 122 | 0.895 [0.41–1.92] | ns (p=0.82) |
| Lung function impairment at 180-day visit | Impaired performance, acute COVID-19, n = 111 | No impaired performance, acute COVID-19, n = 11 | 122 | 0.84 [0.238–3.38] | ns (p=0.82) |
| Lung function impairment at 180-day visit | Sleep disorders, acute COVID-19, n = 46 | No sleep disorders, acute COVID-19, n = 75 | 121 | 0.7 [0.309–1.54] | ns (p=0.44) |
| Lung function impairment at 180-day visit | Anti-infectives, acute COVID-19, n = 63 | No anti-infectives, acute COVID-19, n = 59 | 122 | 2.65 [1.22–6] | p=0.03 |
| Lung function impairment at 180-day visit | Antiplatelet, acute COVID-19, n = 17 | No antiplatelet, acute COVID-19, n = 105 | 122 | 4.8 [1.67–15.1] | p=0.011 |
| Lung function impairment at 180-day visit | Anticoagulatives, acute COVID-19, n = 7 | No anticoagulatives, acute COVID-19, n = 115 | 122 | 2.93 [0.615–15.5] | ns (p=0.23) |
| Lung function impairment at 180-day visit | Immunosuppression, acute COVID-19, n = 22 | No immunosuppression, acute COVID-19, n = 100 | 122 | 2.45 [0.95–6.34] | ns (p=0.096) |
| Lung function impairment at 180-day visit | Anemia, 60-day visit, n = 11 | No anemia, 60-day visit, n = 111 | 122 | 4.14 [1.17–16.7] | ns (p=0.053) |
| Lung function impairment at 180-day visit | Elevated ferritin, 60-day visit, n = 21 | No elevated ferritin, 60-day visit, n = 100 | 121 | 1.37 [0.498–3.6] | ns (p=0.58) |

| Outcome | Covariate | Baseline | Complete cases | OR | pFDR |
|---|---|---|---|---|---|
| Lung function impairment at 180-day visit | Elevated NTproBNP, 60-day visit, n = 44 | No elevated NTproBNP, 60-day visit, n = 78 | 122 | 2.42 [1.11–5.33] | p=0.046 |
| Lung function impairment at 180-day visit | Elevated D-dimer, 60-day visit, n = 50 | No elevated D-dimer, 60-day visit, n = 72 | 122 | 3.23 [1.49–7.2] | p=0.0089 |
| Lung function impairment at 180-day visit | Elevated CRP, 60-day visit, n = 17 | No elevated CRP, 60-day visit, n = 105 | 122 | 6.6 [2.24–22.3] | p=0.0029 |
| Lung function impairment at 180-day visit | Elevated IL-6, 60-day visit, n = 9 | No elevated IL-6, 60-day visit, n = 113 | 122 | 20.2 [3.52–383] | p=0.013 |
| Lung function impairment at 180-day visit | Iron deficiency, 60-day visit, n = 6 | No iron deficiency, 60-day visit, n = 115 | 121 | 1.05 [0.142–5.65] | ns (p=0.96) |
| Lung function impairment at 180-day visit | Age over 65, n = 33 | No age over 65, n = 89 | 122 | 2.55 [1.11–5.88] | p=0.046 |
| Lung function impairment at 180-day visit | Hospitalized >7 days, acute COVID-19, n = 66 | No hospitalized >7 days, acute COVID-19, n = 56 | 122 | 3.83 [1.7–9.21] | p=0.0045 |
| Lung function impairment at 180-day visit | Any comorbidity, n = 93 | No any comorbidity, n = 29 | 122 | 3.95 [1.39–14.2] | p=0.032 |
| Lung function impairment at 180-day visit | >3 comorbidities, n = 41 | No >3 comorbidities, n = 81 | 122 | 2.47 [1.12–5.48] | p=0.044 |
| Lung function impairment at 180-day visit | Overweight or obesity, n = 72 | No overweight or obesity, n = 50 | 122 | 1.24 [0.575–2.73] | ns (p=0.64) |
| Lung function impairment at 180-day visit | >6 symptoms, acute COVID-19, n = 34 | No >6 symptoms, acute COVID-19, n = 88 | 122 | 0.538 [0.207–1.29] | ns (p=0.23) |
| Lung function impairment at 180-day visit | Persistent symptoms, 60-day visit, n = 96 | No persistent symptoms, 60-day visit, n = 26 | 122 | 1.83 [0.702–5.39] | ns (p=0.3) |
| Lung function impairment at 180-day visit | Anti-S1/S2 IgG Q1, 60-day visit, n = 28 | No anti-S1/S2 IgG Q1, 60-day visit, n = 84 | 112 | 0.245 [0.0675–0.704] | p=0.03 |
| Lung function impairment at 180-day visit | Anti-S1/S2 IgG Q2, 60-day visit, n = 27 | No anti-S1/S2 IgG Q2, 60-day visit, n = 85 | 112 | 2.23 [0.913–5.45] | ns (p=0.12) |
| Lung function impairment at 180-day visit | Anti-S1/S2 IgG Q3, 60-day visit, n = 28 | No anti-S1/S2 IgG Q3, 60-day visit, n = 84 | 112 | 0.72 [0.27–1.78] | ns (p=0.55) |
| Lung function impairment at 180-day visit | Anti-S1/S2 IgG Q4, 60-day visit, n = 29 | No anti-S1/S2 IgG Q4, 60-day visit, n = 83 | 112 | 1.88 [0.784–4.51] | ns (p=0.21) |
| Lung function impairment at 180-day visit | Ambulatory, acute COVID-19, n = 32 | No ambulatory, acute COVID-19, n = 90 | 122 | 0.214 [0.0597–0.603] | p=0.017 |
| Lung function impairment at 180-day visit | Hospitalized, acute COVID-19, n = 32 | No hospitalized, acute COVID-19, n = 90 | 122 | 1.1 [0.458–2.56] | ns (p=0.85) |
| Lung function impairment at 180-day visit | Oxygen therapy, acute COVID-19, n = 32 | No oxygen therapy, acute COVID-19, n = 90 | 122 | 1.33 [0.561–3.07] | ns (p=0.57) |
| Lung function impairment at 180-day visit | ICU, acute COVID-19, n = 26 | No ICU, acute COVID-19, n = 96 | 122 | 2.56 [1.05–6.27] | ns (p=0.061) |

| Outcome | Covariate | Baseline | Complete cases | OR | pFDR |
|---|---|---|---|---|---|

CVD = cardiovascular disease; PD = pulmonary disease; COPD = chronic obstructive pulmonary disease; CKD = chronic kidney disease; GITD = gastrointestinal disease; GI = gastrointestinal; CRP = C-reactive protein; ICU = intensive care unit; CT = computed tomography.

**Appendix 1—table 3.** Feature cluster assignment scheme.

| Cluster # | Variable |
|---|---|
| 1 | Male sex, CVD, hypertension, metabolic disorders, anti-infectives, acute COVID-19, elevated NTproBNP, 60-day visit, elevated D-dimer, 60-day visit, hospitalized >7 days, acute COVID-19, >3 comorbidities, overweight |
| 2 | Obesity, current smoker, ex-smoker, PD, COPD, asthma, hypercholesterolemia, diabetes, CKD, GITD, malignancy, immune deficiency, GI symptoms, acute COVID-19, anosmia, acute COVID-19, sleep disorders, acute COVID-19, antiplatelet, acute COVID-19, anticoagulatives, acute COVID-19, immunosuppression, acute COVID-19, anemia, 60-day visit, elevated ferritin, 60-day visit, elevated CRP, 60-day visit, elevated IL-6, 60-day visit, iron deficiency, 60-day visit, age over 65, > 6 symptoms, acute COVID-19, anti-S1/S2 IgG Q1, 60-day visit, anti-S1/S2 IgG Q2, 60-day visit, anti-S1/S2 IgG Q3, 60-day visit, anti-S1/S2 IgG Q4, 60-day visit, ambulatory, acute COVID-19, hospitalized, acute COVID-19, oxygen therapy, acute COVID-19, ICU, acute COVID-19, CT severity score 1–5 at 180-day visit, CT severity score >5 at 180-day visit, lung function impairment at 180-day visit |
| 3 | Weight loss, acute COVID-19, dyspnea, acute COVID-19, cough, acute COVID-19, fever, acute COVID-19, night sweat, acute COVID-19, pain, acute COVID-19, impaired performance, acute COVID-19, any comorbidity, persistent symptoms, 60-day visit, symptoms at 180-day visit |

CVD = cardiovascular disease; PD = pulmonary disease; COPD = chronic obstructive pulmonary disease; CKD = chronic kidney disease; GITD = gastrointestinal disease; GI = gastrointestinal; CRP = C-reactive protein; ICU = intensive care unit; CT = computed tomography.

**Appendix 1—table 4.** Development of machine learning models.
Outcome: outcome variable at the 180-day follow-up visit

| Outcome | Classifier type | Caret method | Description | Package | Optimal arguments |
|---|---|---|---|---|---|
| | | C5.0 | C5.0 | C50 | trials = 10, model = tree, winnow = FALSE |
| | | rf | Random Forest | randomForest | mtry = 27 |
| | | svmRadial | Support Vector Machines with Radial Basis Function Kernel | kernlab | sigma = 0.0105, C = 0.5 |
| | | nnet | Neural Network | nnet | size = 1, decay = 0 |
| | model | glmnet | Elastic-Net Regularized Generalized Linear Models | glmnet | alpha = 0.1, lambda = 0.000431 |
| CT abnormalities at 180-day visit | ensemble | glmnet | Elastic-Net Regularized Generalized Linear Models | glmnet | alpha = 1, lambda = 0.0523 |

*Appendix 1—table 4 Continued on next page*

*Appendix 1—table 4 Continued*

| Outcome | Classifier type | Caret method | Description | Package | Optimal arguments |
|---|---|---|---|---|---|
| | | C5.0 | C5.0 | C50 | trials = 1, model = rules, winnow = TRUE |
| | | rf | Random Forest | randomForest | mtry = 52 |
| | | svmRadial | Support Vector Machines with Radial Basis Function Kernel | kernlab | sigma = 0.00979, C = 0.5 |
| | | nnet | Neural Network | nnet | size = 1, decay = 0.1 |
| | model | glmnet | Elastic-Net Regularized Generalized Linear Models | glmnet | alpha = 0.1, lambda = 0.0419 |
| CT severity score >5 at 180-day visit | ensemble | glmnet | Elastic-Net Regularized Generalized Linear Models | glmnet | alpha = 0.1, lambda = 0.00379 |
| | | C5.0 | C5.0 | C50 | trials = 1, model = tree, winnow = FALSE |
| | | rf | Random Forest | randomForest | mtry = 27 |
| | | svmRadial | Support Vector Machines with Radial Basis Function Kernel | kernlab | sigma = 0.0109, C = 1 |
| | | nnet | Neural Network | nnet | size = 3, decay = 0.1 |
| | model | glmnet | Elastic-Net Regularized Generalized Linear Models | glmnet | alpha = 0.1, lambda = 0.000247 |
| Symptoms at 180-day visit | ensemble | glmnet | Elastic-Net Regularized Generalized Linear Models | glmnet | alpha = 0.1, lambda = 0.0167 |
| | | C5.0 | C5.0 | C50 | trials = 1, model = rules, winnow = FALSE |
| | | rf | Random Forest | randomForest | mtry = 52 |
| | | svmRadial | Support Vector Machines with Radial Basis Function Kernel | kernlab | sigma = 0.0108, C = 0.5 |
| | | nnet | Neural Network | nnet | size = 1, decay = 0.1 |
| | model | glmnet | Elastic-Net Regularized Generalized Linear Models | glmnet | alpha = 0.55, lambda = 0.0341 |
| Lung function impairment at 180-day visit | ensemble | glmnet | Elastic-Net Regularized Generalized Linear Models | glmnet | alpha = 0.55, lambda = 0.0387 |

**Appendix 1—table 5.** Performance of machine learning classifiers.

Outcome: outcome variable at the 180-day follow-up visit; Method: Caret method, Accuracy: model accuracy with 95% confidence intervals, Kappa: model kappa statistic with 95% confidence intervals, AUC: area under the curve.

| Outcome | Total N | Events N | Method | Data set | Accuracy | Kappa | AUC | Sensitivity | Specificity |
|---|---|---|---|---|---|---|---|---|---|
| CT abnormalities at 180-day visit | 109 | 49 | C5.0 | CV | 0.72 [0.36–1] | 0.43 [-0.35–1] | 0.78 | 0.69 | 0.74 |
| CT abnormalities at 180-day visit | 109 | 49 | C5.0 | Training | 1 | 1 | 1 | 1 | 1 |
| CT abnormalities at 180-day visit | 109 | 49 | ensemble | CV | 0.78 [0.63–0.93] | 0.55 [0.26–0.85] | 0.81 | 0.75 | 0.8 |
| CT abnormalities at 180-day visit | 109 | 49 | ensemble | Training | 0.93 | 0.85 | 0.98 | 0.86 | 0.98 |
| CT abnormalities at 180-day visit | 109 | 49 | glmnet | CV | 0.71 [0.3–1] | 0.42 [-0.52–1] | 0.79 | 0.71 | 0.72 |
| CT abnormalities at 180-day visit | 109 | 49 | glmnet | Training | 1 | 1 | 1 | 1 | 1 |
| CT abnormalities at 180-day visit | 109 | 49 | nnet | cCV | 0.67 [0.26–1] | 0.35 [-0.38–1] | 0.69 | 0.71 | 0.64 |
| CT abnormalities at 180-day visit | 109 | 49 | nnet | Training | 0.76 | 0.54 | 0.78 | 1 | 0.57 |
| CT abnormalities at 180-day visit | 109 | 49 | rf | CV | 0.73 [0.4–1] | 0.45 [-0.33–1] | 0.78 | 0.72 | 0.74 |
| CT abnormalities at 180-day visit | 109 | 49 | rf | Training | 1 | 1 | 1 | 1 | 1 |
| CT abnormalities at 180-day visit | 109 | 49 | svmRadial | CV | 0.75 [0.4–1] | 0.51 [-0.25–1] | 0.8 | 0.78 | 0.73 |
| CT abnormalities at 180-day visit | 109 | 49 | svmRadial | Training | 0.85 | 0.7 | 0.93 | 0.84 | 0.87 |
| CT severity score >5 at 180-day visit | 109 | 21 | C5.0 | CV | 0.86 [0.67–1] | 0.37 [-0.2–1] | 0.7 | 0.39 | 0.98 |

*Appendix 1—table 5 Continued on next page*

*Appendix 1—table 5 Continued*

| Outcome | Total N | Events N | Method | Data set | Accuracy | Kappa | AUC | Sensitivity | Specificity |
|---|---|---|---|---|---|---|---|---|---|
| CT severity score >5 at 180-day visit | 109 | 21 | C5.0 | Training | 0.87 | 0.5 | 0.7 | 0.43 | 0.98 |
| CT severity score >5 at 180-day visit | 109 | 21 | ensemble | CV | 0.88 [0.81–0.96] | 0.51 [0.044–0.89] | 0.75 | 0.45 | 0.98 |
| CT severity score >5 at 180-day visit | 109 | 21 | ensemble | Training | 0.89 | 0.57 | 0.65 | 0.48 | 0.99 |
| CT severity score >5 at 180-day visit | 109 | 21 | glmnet | CV | 0.84 [0.6–1] | 0.34 [-0.25–1] | 0.76 | 0.41 | 0.94 |
| CT severity score >5 at 180-day visit | 109 | 21 | glmnet | Training | 0.94 | 0.8 | 0.97 | 0.71 | 1 |
| CT severity score >5 at 180-day visit | 109 | 21 | nnet | CV | 0.79 [0.5–1] | 0.31 [-0.29–1] | 0.72 | 0.47 | 0.87 |
| CT severity score >5 at 180-day visit | 109 | 21 | nnet | Training | 0.99 | 0.97 | 1 | 0.95 | 1 |
| CT severity score >5 at 180-day visit | 109 | 21 | rf | CV | 0.84 [0.6–1] | 0.34 [-0.25–1] | 0.73 | 0.4 | 0.95 |
| CT severity score >5 at 180-day visit | 109 | 21 | rf | Training | 1 | 1 | 1 | 1 | 1 |
| CT severity score >5 at 180-day visit | 109 | 21 | svmRadial | CV | 0.87 [0.63–1] | 0.43 [-0.23–1] | 0.75 | 0.48 | 0.97 |
| CT severity score >5 at 180-day visit | 109 | 21 | svmRadial | Training | 0.92 | 0.68 | 0.99 | 0.57 | 1 |
| Lung function impairment at 180-day visit | 111 | 38 | C5.0 | CV | 0.73 [0.33–1] | 0.39 [-0.5–1] | 0.7 | 0.54 | 0.84 |
| Lung function impairment at 180-day visit | 111 | 38 | C5.0 | Training | 0.86 | 0.7 | 0.85 | 0.79 | 0.9 |
| Lung function impairment at 180-day visit | 111 | 38 | ensemble | CV | 0.75 [0.61–0.86] | 0.39 [0.052–0.67] | 0.72 | 0.48 | 0.89 |

*Appendix 1—table 5 Continued on next page*

| Outcome | Total N | Events N | Method | Data set | Accuracy | Kappa | AUC | Sensitivity | Specificity |
|---|---|---|---|---|---|---|---|---|---|
| Lung function impairment at 180-day visit | 111 | 38 | ensemble | Training | 0.89 | 0.75 | 0.98 | 0.79 | 0.95 |

*Appendix 1—table 5 Continued*

| Outcome | Total N | Events N | Method | Data set | Accuracy | Kappa | AUC | Sensitivity | Specificity |
|---|---|---|---|---|---|---|---|---|---|
| Lung function impairment at 180-day visit | 111 | 38 | glmnet | CV | 0.74 [0.4–1] | 0.37 [-0.36–1] | 0.66 | 0.51 | 0.86 |
| Lung function impairment at 180-day visit | 111 | 38 | glmnet | Training | 0.83 | 0.59 | 0.89 | 0.61 | 0.95 |
| Lung function impairment at 180-day visit | 111 | 38 | nnet | CV | 0.65 [0.2–1] | 0.2 [-0.5–1] | 0.59 | 0.44 | 0.76 |
| Lung function impairment at 180-day visit | 111 | 38 | nnet | Training | 0.93 | 0.83 | 0.82 | 0.79 | 1 |
| Lung function impairment at 180-day visit | 111 | 38 | rf | CV | 0.73 [0.4–1] | 0.35 [-0.33–1] | 0.72 | 0.49 | 0.85 |
| Lung function impairment at 180-day visit | 111 | 38 | rf | Training | 1 | 1 | 1 | 1 | 1 |
| Lung function impairment at 180-day visit | 111 | 38 | svmRadial | CV | 0.72 [0.36–1] | 0.35 [-0.44–1] | 0.69 | 0.5 | 0.84 |
| Lung function impairment at 180-day visit | 111 | 38 | svmRadial | Training | 0.87 | 0.71 | 0.94 | 0.71 | 0.96 |
| Symptoms at 180-day visit | 133 | 65 | C5.0 | CV | 0.6 [0.22–0.93] | 0.2 [-0.51–0.87] | 0.57 | 0.61 | 0.58 |
| Symptoms at 180-day visit | 133 | 65 | C5.0 | Training | 0.93 | 0.86 | 0.96 | 0.89 | 0.97 |
| Symptoms at 180-day visit | 133 | 65 | ensemble | CV | 0.58 [0.41–0.74] | 0.16 [-0.19–0.49] | 0.6 | 0.52 | 0.63 |
| Symptoms at 180-day visit | 133 | 65 | ensemble | Training | 0.99 | 0.98 | 1 | 0.98 | 1 |
| Symptoms at 180-day visit | 133 | 65 | glmnet | CV | 0.56 [0.17–0.86] | 0.13 [-0.64–0.72] | 0.56 | 0.54 | 0.58 |
| Symptoms at 180-day visit | 133 | 65 | glmnet | Training | 0.85 | 0.7 | 0.92 | 0.82 | 0.88 |
| Symptoms at 180-day visit | 133 | 65 | nnet | CV | 0.59 [0.29–0.86] | 0.17 [-0.52–0.72] | 0.58 | 0.6 | 0.57 |
| Symptoms at 180-day visit | 133 | 65 | nnet | Training | 1 | 1 | 1 | 1 | 1 |
| Symptoms at 180-day visit | 133 | 65 | rf | CV | 0.56 [0.29–0.86] | 0.13 [-0.46–0.71] | 0.59 | 0.56 | 0.56 |
| Symptoms at 180-day visit | 133 | 65 | rf | Training | 1 | 1 | 1 | 1 | 1 |
| Symptoms at 180-day visit | 133 | 65 | svmRadial | CV | 0.54 [0.17–0.83] | 0.089 [-0.67–0.67] | 0.55 | 0.45 | 0.62 |
| Symptoms at 180-day visit | 133 | 65 | svmRadial | Training | 0.86 | 0.73 | 0.94 | 0.85 | 0.88 |

AUC = area under the curve; CT = computed tomography; glmnet = elastic-net regularized generalized linear models; nnet = neural networks; svmRadial = support vector machines with radial basis function kernel; rf = random forest; ensemble = model ensemble with elastic-net regularized generalized linear models

**Appendix 1—table 6.** Performance of machine learning classifiers in the acute COVID-19 severity strata.

Outcome: outcome variable at the 180-day follow-up visit; cohort subset: cohort acute COVID-19 severity strata (mild–moderate: outpatient or hospitalized without oxygen; severe–critical: oxygen therapy or ICU),

| Outcome | Cohort subset | Total N | Events N | Method | Data set | AUC | Sensitivity | Specificity |
|---|---|---|---|---|---|---|---|---|
| CT abnormalities at 180-day visit | Whole cohort | 109 | 49 | C5.0 | Training | 1 | 1 | 1 |
| CT abnormalities at 180-day visit | Mild–moderate COVID-19 | 58 | 18 | C5.0 | Training | 1 | 1 | 1 |
| CT abnormalities at 180-day visit | Severe–critical COVID-19 | 51 | 31 | C5.0 | Training | 1 | 1 | 1 |
| CT abnormalities at 180-day visit | Whole cohort | 109 | 49 | rf | Training | 1 | 1 | 1 |
| CT abnormalities at 180-day visit | Mild–moderate COVID-19 | 58 | 18 | rf | Training | 1 | 1 | 1 |
| CT abnormalities at 180-day visit | Severe–critical COVID-19 | 51 | 31 | rf | Training | 1 | 1 | 1 |
| CT abnormalities at 180-day visit | Whole cohort | 109 | 49 | svmRadial | Training | 0.93 | 0.84 | 0.87 |
| CT abnormalities at 180-day visit | Mild–moderate COVID-19 | 58 | 18 | svmRadial | Training | 0.9 | 0.61 | 0.95 |
| CT abnormalities at 180-day visit | Severe–critical COVID-19 | 51 | 31 | svmRadial | Training | 0.96 | 0.97 | 0.7 |
| CT abnormalities at 180-day visit | Whole cohort | 109 | 49 | nnet | Training | 0.78 | 1 | 0.57 |
| CT abnormalities at 180-day visit | Mild–moderate COVID-19 | 58 | 18 | nnet | Training | 0.92 | 1 | 0.85 |
| CT abnormalities at 180-day visit | Severe–critical COVID-19 | 51 | 31 | nnet | Training | 0.5 | 1 | 0 |
| CT abnormalities at 180-day visit | Whole cohort | 109 | 49 | glmnet | Training | 1 | 1 | 1 |
| CT abnormalities at 180-day visit | Mild–moderate COVID-19 | 58 | 18 | glmnet | Training | 1 | 1 | 1 |
| CT abnormalities at 180-day visit | Severe–critical COVID-19 | 51 | 31 | glmnet | Training | 1 | 1 | 1 |
| CT abnormalities at 180-day visit | Whole cohort | 109 | 49 | ensemble | Training | 0.98 | 0.86 | 0.98 |
| CT abnormalities at 180-day visit | Mild–moderate COVID-19 | 58 | 18 | ensemble | Training | 0.98 | 0.61 | 1 |
| CT abnormalities at 180-day visit | Severe–critical COVID-19 | 51 | 31 | ensemble | Training | 1 | 1 | 0.95 |
| CT severity score >5 at 180-day visit | Whole cohort | 109 | 21 | C5.0 | Training | 0.7 | 0.43 | 0.98 |
| CT severity score >5 at 180-day visit | Mild–moderate COVID-19 | 58 | 6 | C5.0 | Training | 0.57 | 0.17 | 0.98 |
| CT severity score >5 at 180-day visit | Severe–critical COVID-19 | 51 | 15 | C5.0 | Training | 0.75 | 0.53 | 0.97 |
| CT severity score >5 at 180-day visit | Whole cohort | 109 | 21 | rf | Training | 1 | 1 | 1 |
| CT severity score >5 at 180-day visit | Mild–moderate COVID-19 | 58 | 6 | rf | Training | 1 | 1 | 1 |
| CT severity score >5 at 180-day visit | Severe–critical COVID-19 | 51 | 15 | rf | Training | 1 | 1 | 1 |

*Appendix 1—table 6 Continued on next page*

Appendix 1—table 6 Continued

| Outcome | Cohort subset | Total N | Events N | Method | Data set | AUC | Sensitivity | Specificity |
|---|---|---|---|---|---|---|---|---|
| CT severity score >5 at 180-day visit | Whole cohort | 109 | 21 | svmRadial | Training | 0.99 | 0.57 | 1 |
| CT severity score >5 at 180-day visit | Mild–moderate COVID-19 | 58 | 6 | svmRadial | Training | 0.98 | 0.17 | 1 |
| CT severity score >5 at 180-day visit | Severe–critical COVID-19 | 51 | 15 | svmRadial | Training | 1 | 0.73 | 1 |
| CT severity score >5 at 180-day visit | Whole cohort | 109 | 21 | nnet | Training | 1 | 0.95 | 1 |
| CT severity score >5 at 180-day visit | Mild–moderate COVID-19 | 58 | 6 | nnet | Training | 1 | 0.83 | 1 |
| CT severity score >5 at 180-day visit | Severe–critical COVID-19 | 51 | 15 | nnet | Training | 1 | 1 | 1 |
| CT severity score >5 at 180-day visit | Whole cohort | 109 | 21 | glmnet | Training | 0.97 | 0.71 | 1 |
| CT severity score >5 at 180-day visit | Mild–moderate COVID-19 | 58 | 6 | glmnet | Training | 0.94 | 0.33 | 1 |
| CT severity score >5 at 180-day visit | Severe–critical COVID-19 | 51 | 15 | glmnet | Training | 1 | 0.87 | 1 |
| CT severity score >5 at 180-day visit | Whole cohort | 109 | 21 | ensemble | Training | 0.65 | 0.48 | 0.99 |
| CT severity score >5 at 180-day visit | Mild–moderate COVID-19 | 58 | 6 | ensemble | Training | 0.38 | 0.17 | 0.98 |
| CT severity score >5 at 180-day visit | Severe–critical COVID-19 | 51 | 15 | ensemble | Training | 0.74 | 0.6 | 1 |
| Symptoms at 180-day visit | Whole cohort | 133 | 65 | C5.0 | Training | 0.96 | 0.89 | 0.97 |
| Symptoms at 180-day visit | Mild–moderate COVID-19 | 64 | 30 | C5.0 | Training | 0.97 | 0.9 | 1 |
| Symptoms at 180-day visit | Severe–critical COVID-19 | 69 | 35 | C5.0 | Training | 0.96 | 0.89 | 0.94 |
| Symptoms at 180-day visit | Whole cohort | 133 | 65 | rf | Training | 1 | 1 | 1 |
| Symptoms at 180-day visit | Mild–moderate COVID-19 | 64 | 30 | rf | Training | 1 | 1 | 1 |
| Symptoms at 180-day visit | Severe–critical COVID-19 | 69 | 35 | rf | Training | 1 | 1 | 1 |
| Symptoms at 180-day visit | Whole cohort | 133 | 65 | svmRadial | Training | 0.94 | 0.85 | 0.88 |
| Symptoms at 180-day visit | Mild–moderate COVID-19 | 64 | 30 | svmRadial | Training | 0.93 | 0.77 | 0.85 |
| Symptoms at 180-day visit | Severe–critical COVID-19 | 69 | 35 | svmRadial | Training | 0.95 | 0.91 | 0.91 |
| Symptoms at 180-day visit | Whole cohort | 133 | 65 | nnet | Training | 1 | 1 | 1 |
| Symptoms at 180-day visit | Mild–moderate COVID-19 | 64 | 30 | nnet | Training | 1 | 1 | 1 |
| Symptoms at 180-day visit | Severe–critical COVID-19 | 69 | 35 | nnet | Training | 1 | 1 | 1 |
| Symptoms at 180-day visit | Whole cohort | 133 | 65 | glmnet | Training | 0.92 | 0.82 | 0.88 |
| Symptoms at 180-day visit | Mild–moderate COVID-19 | 64 | 30 | glmnet | Training | 0.91 | 0.73 | 0.88 |

Appendix 1—table 6 Continued on next page

Appendix 1—table 6 Continued

| Outcome | Cohort subset | Total N | Events N | Method | Data set | AUC | Sensitivity | Specificity |
|---|---|---|---|---|---|---|---|---|
| Symptoms at 180-day visit | Severe–critical COVID-19 | 69 | 35 | glmnet | Training | 0.92 | 0.89 | 0.88 |
| Symptoms at 180-day visit | Whole cohort | 133 | 65 | ensemble | Training | 1 | 0.98 | 1 |
| Symptoms at 180-day visit | Mild–moderate COVID-19 | 64 | 30 | ensemble | Training | 1 | 0.97 | 1 |
| Symptoms at 180-day visit | Severe–critical COVID-19 | 69 | 35 | ensemble | Training | 1 | 1 | 1 |
| Lung function impairment at 180-day visit | Whole cohort | 111 | 38 | C5.0 | Training | 0.85 | 0.79 | 0.9 |
| Lung function impairment at 180-day visit | Mild–moderate COVID-19 | 55 | 14 | C5.0 | Training | 0.81 | 0.71 | 0.9 |
| Lung function impairment at 180-day visit | Severe–critical COVID-19 | 56 | 24 | C5.0 | Training | 0.87 | 0.83 | 0.91 |
| Lung function impairment at 180-day visit | Whole cohort | 111 | 38 | rf | Training | 1 | 1 | 1 |
| Lung function impairment at 180-day visit | Mild–moderate COVID-19 | 55 | 14 | rf | Training | 1 | 1 | 1 |
| Lung function impairment at 180-day visit | Severe–critical COVID-19 | 56 | 24 | rf | Training | 1 | 1 | 1 |
| Lung function impairment at 180-day visit | Whole cohort | 111 | 38 | svmRadial | Training | 0.94 | 0.71 | 0.96 |
| Lung function impairment at 180-day visit | Mild–moderate COVID-19 | 55 | 14 | svmRadial | Training | 0.88 | 0.5 | 0.98 |
| Lung function impairment at 180-day visit | Severe–critical COVID-19 | 56 | 24 | svmRadial | Training | 0.98 | 0.83 | 0.94 |
| Lung function impairment at 180-day visit | Whole cohort | 111 | 38 | nnet | Training | 0.82 | 0.79 | 1 |
| Lung function impairment at 180-day visit | Mild–moderate COVID-19 | 55 | 14 | nnet | Training | 0.7 | 0.64 | 1 |
| Lung function impairment at 180-day visit | Severe–critical COVID-19 | 56 | 24 | nnet | Training | 0.89 | 0.88 | 1 |
| Lung function impairment at 180-day visit | Whole cohort | 111 | 38 | glmnet | Training | 0.89 | 0.61 | 0.95 |
| Lung function impairment at 180-day visit | Mild–moderate COVID-19 | 55 | 14 | glmnet | Training | 0.84 | 0.29 | 0.95 |
| Lung function impairment at 180-day visit | Severe–critical COVID-19 | 56 | 24 | glmnet | Training | 0.91 | 0.79 | 0.94 |
| Lung function impairment at 180-day visit | Whole cohort | 111 | 38 | ensemble | Training | 0.98 | 0.79 | 0.95 |

| Outcome | Cohort subset | Total N | Events N | Method | Data set | AUC | Sensitivity | Specificity |
|---|---|---|---|---|---|---|---|---|
| Lung function impairment at 180-day visit | Mild–moderate COVID-19 | 55 | 14 | ensemble | Training | 0.97 | 0.71 | 0.95 |
| Lung function impairment at 180-day visit | Severe–critical COVID-19 | 56 | 24 | ensemble | Training | 0.98 | 0.83 | 0.94 |
| CT abnormalities at 180-day visit | Whole cohort | 109 | 49 | C5.0 | CV | 0.78 | 0.69 | 0.74 |
| CT abnormalities at 180-day visit | Mild–moderate COVID-19 | 58 | 18 | C5.0 | CV | 0.69 | 0.43 | 0.8 |
| CT abnormalities at 180-day visit | Severe–critical COVID-19 | 51 | 31 | C5.0 | CV | 0.78 | 0.85 | 0.62 |
| CT abnormalities at 180-day visit | Whole cohort | 109 | 49 | rf | CV | 0.78 | 0.72 | 0.74 |
| CT abnormalities at 180-day visit | Mild–moderate COVID-19 | 58 | 18 | rf | CV | 0.76 | 0.43 | 0.88 |
| CT abnormalities at 180-day visit | Severe–critical COVID-19 | 51 | 31 | rf | CV | 0.71 | 0.88 | 0.47 |
| CT abnormalities at 180-day visit | Whole cohort | 109 | 49 | svmRadial | CV | 0.8 | 0.78 | 0.73 |
| CT abnormalities at 180-day visit | Mild–moderate COVID-19 | 58 | 18 | svmRadial | CV | 0.75 | 0.56 | 0.9 |
| CT abnormalities at 180-day visit | Severe–critical COVID-19 | 51 | 31 | svmRadial | CV | 0.76 | 0.92 | 0.4 |
| CT abnormalities at 180-day visit | Whole cohort | 109 | 49 | nnet | CV | 0.69 | 0.71 | 0.64 |
| CT abnormalities at 180-day visit | Mild–moderate COVID-19 | 58 | 18 | nnet | CV | 0.67 | 0.59 | 0.77 |
| CT abnormalities at 180-day visit | Severe–critical COVID-19 | 51 | 31 | nnet | CV | 0.62 | 0.78 | 0.39 |
| CT abnormalities at 180-day visit | Whole cohort | 109 | 49 | glmnet | CV | 0.79 | 0.71 | 0.72 |
| CT abnormalities at 180-day visit | Mild–moderate COVID-19 | 58 | 18 | glmnet | CV | 0.78 | 0.66 | 0.78 |
| CT abnormalities at 180-day visit | Severe–critical COVID-19 | 51 | 31 | glmnet | CV | 0.75 | 0.75 | 0.6 |
| CT abnormalities at 180-day visit | Whole cohort | 109 | 49 | ensemble | CV | 0.81 | 0.75 | 0.8 |
| CT abnormalities at 180-day visit | Mild–moderate COVID-19 | 58 | 18 | ensemble | CV | 0.76 | 0.55 | 0.92 |
| CT abnormalities at 180-day visit | Severe–critical COVID-19 | 51 | 31 | ensemble | CV | 0.79 | 0.87 | 0.55 |
| CT severity score >5 at 180-day visit | Whole cohort | 109 | 21 | C5.0 | CV | 0.7 | 0.39 | 0.98 |
| CT severity score >5 at 180-day visit | Mild–moderate COVID-19 | 58 | 6 | C5.0 | CV | 0.55 | 0.13 | 0.98 |
| CT severity score >5 at 180-day visit | Severe–critical COVID-19 | 51 | 15 | C5.0 | CV | 0.76 | 0.49 | 0.97 |
| CT severity score >5 at 180-day visit | Whole cohort | 109 | 21 | rf | CV | 0.73 | 0.4 | 0.95 |
| CT severity score >5 at 180-day visit | Mild–moderate COVID-19 | 58 | 6 | rf | CV | 0.58 | 0.033 | 0.96 |
| CT severity score >5 at 180-day visit | Severe–critical COVID-19 | 51 | 15 | rf | CV | 0.76 | 0.55 | 0.93 |

| Outcome | Cohort subset | Total N | Events N | Method | Data set | AUC | Sensitivity | Specificity |
|---|---|---|---|---|---|---|---|---|
| CT severity score >5 at 180-day visit | Whole cohort | 109 | 21 | svmRadial | CV | 0.75 | 0.48 | 0.97 |
| CT severity score >5 at 180-day visit | Mild–moderate COVID-19 | 58 | 6 | svmRadial | CV | 0.59 | 0.13 | 0.97 |
| CT severity score >5 at 180-day visit | Severe–critical COVID-19 | 51 | 15 | svmRadial | CV | 0.79 | 0.61 | 0.97 |
| CT severity score >5 at 180-day visit | Whole cohort | 109 | 21 | nnet | CV | 0.72 | 0.47 | 0.87 |
| CT severity score >5 at 180-day visit | Mild–moderate COVID-19 | 58 | 6 | nnet | CV | 0.57 | 0.17 | 0.89 |
| CT severity score >5 at 180-day visit | Severe–critical COVID-19 | 51 | 15 | nnet | CV | 0.76 | 0.59 | 0.84 |
| CT severity score >5 at 180-day visit | Whole cohort | 109 | 21 | glmnet | CV | 0.76 | 0.41 | 0.94 |
| CT severity score >5 at 180-day visit | Mild–moderate COVID-19 | 58 | 6 | glmnet | CV | 0.63 | 0.17 | 0.97 |
| CT severity score >5 at 180-day visit | Severe–critical COVID-19 | 51 | 15 | glmnet | CV | 0.78 | 0.51 | 0.89 |
| CT severity score >5 at 180-day visit | Whole cohort | 109 | 21 | ensemble | CV | 0.75 | 0.45 | 0.98 |
| CT severity score >5 at 180-day visit | Mild–moderate COVID-19 | 58 | 6 | ensemble | CV | 0.64 | 0.17 | 0.98 |
| CT severity score >5 at 180-day visit | Severe–critical COVID-19 | 51 | 15 | ensemble | CV | 0.78 | 0.57 | 0.99 |
| Symptoms at 180-day visit | Whole cohort | 133 | 65 | C5.0 | CV | 0.57 | 0.61 | 0.58 |
| Symptoms at 180-day visit | Mild–moderate COVID-19 | 64 | 30 | C5.0 | CV | 0.58 | 0.62 | 0.56 |
| Symptoms at 180-day visit | Severe–critical COVID-19 | 69 | 35 | C5.0 | CV | 0.55 | 0.6 | 0.6 |
| Symptoms at 180-day visit | Whole cohort | 133 | 65 | rf | CV | 0.59 | 0.56 | 0.56 |
| Symptoms at 180-day visit | Mild–moderate COVID-19 | 64 | 30 | rf | CV | 0.6 | 0.61 | 0.55 |
| Symptoms at 180-day visit | Severe–critical COVID-19 | 69 | 35 | rf | CV | 0.57 | 0.52 | 0.58 |
| Symptoms at 180-day visit | Whole cohort | 133 | 65 | svmRadial | CV | 0.55 | 0.45 | 0.62 |
| Symptoms at 180-day visit | Mild–moderate COVID-19 | 64 | 30 | svmRadial | CV | 0.54 | 0.48 | 0.59 |
| Symptoms at 180-day visit | Severe–critical COVID-19 | 69 | 35 | svmRadial | CV | 0.56 | 0.43 | 0.66 |
| Symptoms at 180-day visit | Whole cohort | 133 | 65 | nnet | CV | 0.58 | 0.6 | 0.57 |
| Symptoms at 180-day visit | Mild–moderate COVID-19 | 64 | 30 | nnet | CV | 0.58 | 0.63 | 0.54 |
| Symptoms at 180-day visit | Severe–critical COVID-19 | 69 | 35 | nnet | CV | 0.58 | 0.58 | 0.6 |
| Symptoms at 180-day visit | Whole cohort | 133 | 65 | glmnet | CV | 0.56 | 0.54 | 0.58 |
| Symptoms at 180-day visit | Mild–moderate COVID-19 | 64 | 30 | glmnet | CV | 0.56 | 0.57 | 0.6 |
| Symptoms at 180-day visit | Severe–critical COVID-19 | 69 | 35 | glmnet | CV | 0.55 | 0.51 | 0.56 |

| Outcome | Cohort subset | Total N | Events N | Method | Data set | AUC | Sensitivity | Specificity |
|---|---|---|---|---|---|---|---|---|
| Symptoms at 180-day visit | Whole cohort | 133 | 65 | ensemble | CV | 0.6 | 0.52 | 0.63 |
| Symptoms at 180-day visit | Mild–moderate COVID-19 | 64 | 30 | ensemble | CV | 0.61 | 0.57 | 0.63 |
| Symptoms at 180-day visit | Severe–critical COVID-19 | 69 | 35 | ensemble | CV | 0.6 | 0.49 | 0.63 |
| Lung function impairment at 180-day visit | Whole cohort | 111 | 38 | C5.0 | CV | 0.7 | 0.54 | 0.84 |
| Lung function impairment at 180-day visit | Mild–moderate COVID-19 | 55 | 14 | C5.0 | CV | 0.61 | 0.37 | 0.86 |
| Lung function impairment at 180-day visit | Severe–critical COVID-19 | 56 | 24 | C5.0 | CV | 0.75 | 0.63 | 0.81 |
| Lung function impairment at 180-day visit | Whole cohort | 111 | 38 | rf | CV | 0.72 | 0.49 | 0.85 |
| Lung function impairment at 180-day visit | Mild–moderate COVID-19 | 55 | 14 | rf | CV | 0.58 | 0.26 | 0.9 |
| Lung function impairment at 180-day visit | Severe–critical COVID-19 | 56 | 24 | rf | CV | 0.79 | 0.62 | 0.79 |
| Lung function impairment at 180-day visit | Whole cohort | 111 | 38 | svmRadial | CV | 0.69 | 0.5 | 0.84 |
| Lung function impairment at 180-day visit | Mild–moderate COVID-19 | 55 | 14 | svmRadial | CV | 0.56 | 0.29 | 0.88 |
| Lung function impairment at 180-day visit | Severe–critical COVID-19 | 56 | 24 | svmRadial | CV | 0.75 | 0.62 | 0.79 |
| Lung function impairment at 180-day visit | Whole cohort | 111 | 38 | nnet | CV | 0.59 | 0.44 | 0.76 |
| Lung function impairment at 180-day visit | Mild–moderate COVID-19 | 55 | 14 | nnet | CV | 0.47 | 0.29 | 0.83 |
| Lung function impairment at 180-day visit | Severe–critical COVID-19 | 56 | 24 | nnet | CV | 0.62 | 0.52 | 0.67 |
| Lung function impairment at 180-day visit | Whole cohort | 111 | 38 | glmnet | CV | 0.66 | 0.51 | 0.86 |
| Lung function impairment at 180-day visit | Mild–moderate COVID-19 | 55 | 14 | glmnet | CV | 0.55 | 0.21 | 0.88 |
| Lung function impairment at 180-day visit | Severe–critical COVID-19 | 56 | 24 | glmnet | CV | 0.73 | 0.68 | 0.83 |
| Lung function impairment at 180-day visit | Whole cohort | 111 | 38 | ensemble | CV | 0.72 | 0.48 | 0.89 |
| Lung function impairment at 180-day visit | Mild–moderate COVID-19 | 55 | 14 | ensemble | CV | 0.59 | 0.26 | 0.92 |
| Lung function impairment at 180-day visit | Severe–critical COVID-19 | 56 | 24 | ensemble | CV | 0.78 | 0.61 | 0.85 |

| Outcome | Cohort subset | Total N | Events N | Method | Data set | AUC | Sensitivity | Specificity |
|---------|---------------|---------|----------|--------|----------|-----|-------------|-------------|

AUC = area under the curve; CT = computed tomography; ICU = intensive care unit; glmnet = elastic-net regularized generalized linear models; nnet = neural network; svmRadial = support vector machines with radial basis function kernel; rf = random forest; ensemble = model ensemble with elastic-net regularized generalized linear models

