## [Editor Report]

This is an informative paper describing the incidence and predictors of long-term radiological and functional lung abnormalities following COVID-19. Congratulations on the importance of the work!

---

## [Decision Letter]

**Decision letter after peer review:**

Thank you for submitting your article "Investigating phenotypes of pulmonary COVID- 19 recovery -a longitudinal observational prospective multicenter trial" for consideration by *eLife*. Your article has been reviewed by 2 peer reviewers, including Joshua T Schiffer as the Reviewing Editor and Reviewer #2, and the evaluation has been overseen by Jos Van der Meer as the Senior Editor. The following individual involved in review of your submission has agreed to reveal their identity: Guang-Shing Cheng (Reviewer #1).

Essential revisions:

1) Please describe potential methods to operationalize the machine learning approach.

2) Please take care to precisely define all endpoints throughout the study as per the requests of reviewer 3.

3) Please provide rationale for using radiologic endpoints rather than clinical and functional endpoints throughout the study and consider further analyses using clinical and functional endpoints.

4) Please clearly discriminate the radiologic endpoints. If severe abnormalities of a subset of any abnormality, then please explicitly state this. It is necessary to include the number of study participants who meet each endpoint to provide complete clarity.

5) More precisely describe the meaning of low, medium and high risk in figure 6.

6) Please be sure that exposure and outcome variables are independent of one another or remove the exposure variable from the analysis.

*Reviewer #1 (Recommendations for the authors):*

Well done study, well written, and of great interest to me personally and scientifically.

1. Would you be able to apply your machine learning algorithms to an external validation cohort from the same time frame? Would lend additional support to your model.

2. Additional follow-up assessments at 1 year would be informative, but perhaps that data is forthcoming in another manuscript.

3. How would you operationalize ML algorithms for clinical use?

*Reviewer #2 (Recommendations for the authors):*

1) Please justify selection of radiologic endpoints as primary endpoints rather than functional and symptomatic endpoints.

2) Please define endpoints specifically and explicitly state the number of patients who fall within each endpoint, taking great care to discriminate whether different groups overlap.

---

## [Author Response]

Essential revisions:1) Please describe potential methods to operationalize the machine learning approach.

We appreciate this important point. The applicability of a machine learning algorithm to classify real-life data is the central challenge and the greatest strength of the approach. Unfortunately, our longitudinal one-cohort study does not provide us with a possibility of external validation of the clustering and classification procedures presented in the manuscript. For this reason, we hesitated to discuss extensively the performance features and the reproducibility of the presented machine learning algorithms in the initial manuscript.

In the revised manuscript we discuss the potential of machine-learning-assisted analysis of medical record data, laboratory and patient self-reported data in early prediction of COVID-19 severity [1–3] as well as prediction and phenotyping of complicated recovery [3–6]. In addition to gain more confidence in the robustness of the clustering and classification procedures shown in the manuscript, we consistently included the 20-fold cross-validation for all those analyses instead of the repeated holdout strategy used previously (feature cluster validation: Figure 6—figure supplement 1, participant cluster validation: Figure 7—figure supplement 1, validation of machine learning models: Figure 9 and Appendix 1 – table 5).

Finally, we developed an online, open source pulmonary assessment tools based on the R Shiny platform (https://im2-ibk.shinyapps.io/CovILD/, code available from https://github.com/PiotrTymoszuk/COVILD-recovery-assessment-app). The tool implements assignment of the user-provided patient records to the Risk Clusters described in the manuscript. In addition, it enables predictions of any lung CT abnormalities, moderate-to-severe CT abnormalities or functional lung impairment at the 180-day follow-up with the machine learning algorithms presented in the manuscript, which were trained and cross-validated in the CovILD cohort. We believe that such tool can increase the visibility of our work, foster collaboration, and give us an opportunity to validate our clustering approach in the future.

2) Please take care to precisely define all endpoints throughout the study as per the requests of reviewer 3.

We are thankful for pointing out this unclarity. In the revised manuscript, we precisely define the primary (any radiological lung findings at the 6-month follow-up) and secondary endpoints (radiological lung abnormalities with CT score > 5, lung function impairment and persistent symptoms at the 6-month follow-up) of the study and analysis. See: Introduction and Methods/Study design for the description in the text and Table 3 with the numbers and percentages of the study participants reaching the endpoints. The overlap between the subjects reaching the radiological, functional and clinical endpoints is presented in Figure 3—figure supplement 1 and Figure 3—figure supplement 2.

3) Please provide rationale for using radiologic endpoints rather than clinical and functional endpoints throughout the study and consider further analyses using clinical and functional endpoints.

This is an important issue which we clarify in the revised manuscript. First, the study was established after the emergence of the first COVID-19 in Europa in March 2020, and at this time hardly any information was available concerning the pulmonary outcome of COVID-19. One major issue was the concern that comparable to SARS-CoV-1 infection, many patients may develop long-term persistent structural lung abnormalities in general and interstitial lung disease (ILD) in particular following acute COVID-19 pneumonia [7–9]. Thus, we implemented computed tomography as a primary assessment tool, which is the best diagnostic tool to assess early ILD [10,11].

Another goal of the CovILD study was to provide evidence for the development of structured follow-up algorithms for COVID-19 patients. As medical resources are limited, especially during a pandemic, we aimed to identify surrogate parameters, which enable us to identify patients at risk for structural pulmonary damage and the need for close-meshed functional and radiological follow-up. In this context, clinical symptoms, which are typically multifactorial and do not necessarily aid early identification of ILD, were used as a secondary outcome parameter.

Still, we agree, that clinical and functional endpoints are of great interest for the scientific, clinical and patient community. For this reason, we additionally included the long-term symptom persistence and lung function impairment outcome variables in the univariate (Figure 5, Appendix 1 – table 2) and machine learning multi-parameter risk modeling (Figure 9 – 10, Appendix 1 – table 5 and Appendix 1 – table 6). We also compare the frequency of those outcome variables in the Risk Clusters of the study participants (Figure 8).

4) Please clearly discriminate the radiologic endpoints. If severe abnormalities of a subset of any abnormality, then please explicitly state this. It is necessary to include the number of study participants who meet each endpoint to provide complete clarity.

We now clarify this important point in the Introduction, Methods and Results. N numbers of the participants meeting the endpoints at subsequent follow-up visits are presented in Table 2.

The individuals with CT severity score > 5 were a subset of the participants with any CT abnormality. The same was true for the GGO-positive patients. We agree with Editor and Reviewer 2, that the overlap between the radiological outcomes obscures the message of the clustering and modeling results. To overcome this, we removed the GGO outcome variable from the kinetic (Figure 3) and risk modeling (Figure 4), machine learning classification (Figure 9 and 10) and comparisons of CT abnormality frequency between the patient clusters (Figure 8). Please note that the great majority of the CT findings present in the study collective was anyway classified as GGOs and a detailed characteristic of CT abnormalities will be addressed by another report of our study team (Luger A et al., in revision).

In the revised manuscript, we differentiate between mild (CT severity score ≤ 5) and moderate-to-severe radiological abnormalities (CT severity score > 5) in feature (Figure 6) and participant clustering (Figure 8). Furthermore, to guarantee the consequent distinction of explanatory and outcome variables, both the feature (Figure 6) and participant clusters (Figure 7) are defined exclusively with non-CT variables. To investigate the association of mild and moderate-to-severe CT abnormalities with other non-CT variables (Figure 6), the CT features are assigned to the no-CT clusters by a k-NN-based label propagation algorithm, i. e. semi-supervised procedure [12,13] employed in our recent paper as well [6].

5) More precisely describe the meaning of low, medium and high risk in figure 6.

The nomenclature: low-, intermediate- and high- risk pertains to the frequency of long-term radiological lung abnormalities in the study participant clusters (Figure 8A). We now describe it more clearly in the section ‘Results/Risk stratification for perturbed pulmonary recovery by unsupervised clustering’ before describing other cluster features.

6) Please be sure that exposure and outcome variables are independent of one another or remove the exposure variable from the analysis.

This is an important issue. We agree with the argumentation of the Editor and Reviewer 3 that the inclusion of the overlapping CT responses in risk modeling, clustering and machine learning classification obfuscates the conclusions. For this reason and the reasons described in response to Essential revisions comment 4, we removed the GGO variable from the revised analysis pipeline and differentiate between mild (CT severity score ≤ 5) and moderate-to-severe (CT severity score > 5) radiological lung abnormalities in the modeling, clustering and machine learning classification. In addition, we define symptom and participant clusters solely with the non-CT parameters. See response to Essential revisions comment 4 for details.

Reviewer #1 (Recommendations for the authors):Well done study, well written, and of great interest to me personally and scientifically.1. Would you be able to apply your machine learning algorithms to an external validation cohort from the same time frame? Would lend additional support to your model.

This is an extremely important point. As stressed already in the initial version of the manuscript, the optimal performance of clustering and classification algorithms can be achieved with large training cohorts and external validation is a crucial step, which would require an additional cohort with a comparable parameter record. We searched for external collaborators, but unfortunately, none of the contacted collaborating academic pulmonology centers in Europe could provide us with a comparably rich set of demography, clinics, biochemistry, functional and imaging data collected at analogical time points. Furthermore, large cross-sectional observation cohorts, like the Wuhan Study [14], do not offer open data access.

In the revised manuscript we tried to improve the robustness of the classifiers and partially address the lacking possibility for external validation:

1. We do not restrict the analysis to the subset of the CovILD study with the complete set of all variables. Instead, the non-missingness criterion is applied to each outcome variable separately (any CT abnormalities: n = 109, moderate-to-severe abnormalities: n = 109, lung function impairment: n = 111, persistent symptoms: n = 133). This resulted in a greater number of observations used for training of the machine learning algorithms.

2. We altered the internal validation strategy. Instead of the repeated holdout approach applied to the machine learning classification, which strongly limits the size of the training data set, we switched to 20-fold cross-validation both for the cluster algorithms (as described by Lange et al. [15], Figure 6—figure supplement 1BD and Figure 7—figure supplement 1BF) and the machine learning models (Figure 9, Appendix 1 – table 5).

3. The algorithm of clustering of the study participants was changed to a more stable one as investigated by 20-fold cross -validation stability test [15]. Instead of the k-means procedure [16] applied to the clinical non-CT parameters in the initial manuscript version, a combined self-organizing map (SOM) – hierarchical clustering algorithm is used (Supplementary Figure S6) [17,18]. Importantly, both methods classified the study participants in a comparable way into clusters differing significantly in frequency of pulmonary CT abnormalities (Figure 8). In addition, the cluster assignment was shown to be a significant correlate of persistent radiological lung abnormalities independently of the acute COVID-19 severity (Figure 8—figure supplement 1B), as shown in the initial version of the manuscript.

4. The set of machine learning models were optimized and includes now multiple tools provided by the R package *caret* [19]*,* which represent various families of machine learning algorithms (rule tree classifier: C5.0 [20], bagged tree classifier: Random Forests [21], support vector machines (SVM) with radial kernel [22], shallow neural networks: nnet [23] and elastic net: glmnet [24], Appendix 1 – table 4) and provided more consistent results than the simple kNN and naive Bayes algorithms presented before. Finally, ensemble models being a linear combination of the C5.0, Radom Forests, SVM, nnet and glmnet classifiers were constructed based on the elastic net algorithm and *caretEnsemble* package (Figure 9—figure supplement 2) [25]. Notably, the respective ensemble models showed a superior accuracy at predicting any CT abnormalities and persistent symptoms in the cross-validation setting (Figure 9, Appendix 1 – table 5).

5. We provide an open-source online R Shiny application (https://im2-ibk.shinyapps.io/CovILD/) implementing k-NN-based assignment (supervised clustering) [12,13,26] the user-provided patient’s records to the low-, intermediate- and high-risk clusters (Figure 7) and enables pulmonary outcome prediction by machine learning. We suppose that such tool may foster collaboration and facilitate verification of the manuscript’s findings intramurally and by all interested centers.

2. Additional follow-up assessments at 1 year would be informative, but perhaps that data is forthcoming in another manuscript.

Unfortunately, we are not able to disclose the one-year follow-up data in the revised manuscript, as the radiological and clinical findings of the CovILD cohort are included in a manuscript currently under revision in *Radiology* (Luger A et al.) and we are still working on the analysis of the clinical, cardiopulmonary and mental recovery data for the one-year follow-up time point.

3. How would you operationalize ML algorithms for clinical use?

We appreciate this point. The drawback of the clustering and classification algorithms presented in the manuscript is the large set of input variables (50 non-CT features) precluding manual one-by-one risk computation in clinical routine. As described in response to the public review, an open-source risk cluster classification and machine learning pulmonary outcome prediction tool accompanies now the revised manuscript (https://im2-ibk.shinyapps.io/CovILD/). Such tool implements a data sheet (.xlsx) input, enabling concomitant analysis of multiple patient records at a time.

Reviewer #2 (Recommendations for the authors):1) Please justify selection of radiologic endpoints as primary endpoints rather than functional and symptomatic endpoints.

Please see response to the public review for the motivation of the focus on radiological/structural lung recovery.

2) Please define endpoints specifically and explicitly state the number of patients who fall within each endpoint, taking great care to discriminate whether different groups overlap.

Done as requested, please see response to the public review and Table 3 of the revised manuscript.

References:

1. Gutmann C, Takov K, Burnap SA, et al. SARS-CoV-2 RNAemia and proteomic trajectories inform prognostication in COVID-19 patients admitted to intensive care. Nat Commun 2021;12. doi:10.1038/S41467-021-23494-1

2. Benito-León J, Castillo MD Del, Estirado A, et al. Using Unsupervised Machine Learning to Identify Age- and Sex-Independent Severity Subgroups Among Patients with COVID-19: Observational Longitudinal Study. J Med Internet Res 2021;23. doi:10.2196/25988

3. Demichev V, Tober-Lau P, Lemke O, et al. A time-resolved proteomic and prognostic map of COVID-19. Cell Syst 2021;12:780. doi:10.1016/J.CELS.2021.05.005

4. Estiri H, Strasser ZH, Brat GA, et al. Evolving phenotypes of non-hospitalized patients that indicate long COVID. BMC Med 2021;19. doi:10.1186/S12916-021-02115-0

5. Sudre CH, Murray B, Varsavsky T, et al. Attributes and predictors of long COVID. Nat Med 2021;27. doi:10.1038/s41591-021-01292-y

6. Sahanic S, Tymoszuk P, Ausserhofer D, et al. Phenotyping of acute and persistent COVID-19 features in the outpatient setting: exploratory analysis of an international cross-sectional online survey. Clin Infect Dis Published Online First: 26 November 2021. doi:10.1093/CID/CIAB978

7. Hui DS, Wong KT, Ko FW, et al. The 1-Year Impact of Severe Acute Respiratory Syndrome on Pulmonary Function, Exercise Capacity, and Quality of Life in a Cohort of Survivors. Chest 2005;128:2247–61. doi:10.1378/CHEST.128.4.2247

8. Ng CK, Chan JWM, Kwan TL, et al. Six month radiological and physiological outcomes in severe acute respiratory syndrome (SARS) survivors. Thorax 2004;59:889–91. doi:10.1136/THX.2004.023762

9. Raghu G, Wilson KC. COVID-19 interstitial pneumonia: monitoring the clinical course in survivors. Lancet Respir. Med. 2020;8:839–42. doi:10.1016/S2213-2600(20)30349-0

10. Suliman YA, Dobrota R, Huscher D, et al. Pulmonary function tests: High rate of false-negative results in the early detection and screening of scleroderma-related interstitial lung disease. Arthritis Rheumatol 2015;67:3256–61. doi:10.1002/ART.39405/ABSTRACT

11. Hatabu H, Hunninghake GM, Richeldi L, et al. Interstitial lung abnormalities detected incidentally on CT: a Position Paper from the Fleischner Society. Lancet Respir Med 2020;8:726. doi:10.1016/S2213-2600(20)30168-5

12. Leng M, Wang J, Cheng J, et al. Adaptive semi-supervised clustering algorithm with label propagation. J Softw Eng 2014;8:14–22. doi:10.3923/JSE.2014.14.22

13. Lelis L, Sander J. Semi-supervised density-based clustering. Proc – IEEE Int Conf Data Mining, ICDM 2009;:842–7. doi:10.1109/ICDM.2009.143

14. Huang C, Huang L, Wang Y, et al. 6-month consequences of COVID-19 in patients discharged from hospital: a cohort study. Lancet 2021;397:220–32. doi:10.1016/S0140-6736(20)32656-8

15. Lange T, Roth V, Braun ML, et al. Stability-Based Validation of Clustering Solutions. Neural Comput 2004;16:1299–323. doi:10.1162/089976604773717621

16. Hartigan JA, Wong MA. Algorithm AS 136: A K-Means Clustering Algorithm. Appl Stat 1979;28:100. doi:10.2307/2346830

17. Kohonen T. Self-Organizing Maps. Berlin, Heidelberg: : Springer Berlin Heidelberg 1995. doi:10.1007/978-3-642-97610-0

18. Vesanto J, Alhoniemi E. Clustering of the self-organizing map. IEEE Trans Neural Networks 2000;11:586–600. doi:10.1109/72.846731

19. Kuhn M. Building predictive models in R using the caret package. J Stat Softw 2008;28:1–26. doi:10.18637/jss.v028.i05

20. Quinlan JR. C4.5: Programs for Machine Learning. San Francisco, CA, USA: : Morgan Kaufmann Publishers Inc. 1993. doi:10.5555/152181

21. Breiman L. Random forests. Mach Learn 2001;45:5–32. doi:10.1023/A:1010933404324

22. Weston J, Watkins C. Multi-Class Support Vector Machines. 1998.

23. Ripley BD. Pattern recognition and neural networks. Cambridge University Press 2014. doi:10.1017/CBO9780511812651

24. Friedman J, Hastie T, Tibshirani R. Regularization paths for generalized linear models via coordinate descent. J Stat Softw 2010;33:1–22. doi:10.18637/jss.v033.i01

25. Deane-Mayer ZA, Knowles JE. Ensembles of Caret Models [R package caretEnsemble version 2.0.1]. 2019.https://cran.r-project.org/package=caretEnsemble (accessed 13 Dec 2021).

26. Glennan T, Leckie C, Erfani SM. Improved Classification of Known and Unknown Network Traffic Flows Using Semi-supervised Machine Learning. Lect Notes Comput Sci (including Subser Lect Notes Artif Intell Lect Notes Bioinformatics) 2016;9723:493–501. doi:10.1007/978-3-319-40367-0_33

27. Sonnweber T, Sahanic S, Pizzini A, et al. Cardiopulmonary recovery after COVID-19 – an observational prospective multi-center trial. Eur Respir J Published Online First: 10 December 2020. doi:10.1183/13993003.03481-2020